# Delving into Non-Exchangeability for Conformal Prediction in Graph-Structured Multivariate Time Series

**Ruichao Guo** [* 1]  **Xingyao Han** [* 1]  **Wenshui Luo** [1]  **Zhe Liu** [1]  **Chen Gong** [1]  **Hesheng Wang** [1]

## Abstract

Point forecasting for graph-structured multivariate time series is a fundamental problem, but rigorous uncertainty quantification for such predictions is still underexplored. Conformal prediction (CP) offers uncertainty estimation with a solid coverage guarantee under the exchangeability assumption, which requires the joint data distribution to be unchanged under permutation. However, in graph-structured time series, inherent cross-node coupling can violate the exchangeability condition, making direct application of CP unreliable. Inspired by the spectral graph theory, such coupling resides in global trends and can be characterized by the low-frequency components, while high-frequency components are nearly exchangeable. Therefore, we propose a novel concept named **S**pectral **G**raph **C**onditional **E**xchangeability (SGCE), which conditions exchangeable high-frequency components on low-frequency ones to preserve global trends and enable effective CP in the spectral domain. Based on SGCE, we further propose **S**pectral **C**onformal prediction via w**A**ve**LE**t transform (SCALE). SCALE uses graph wavelets to decompose low/high-frequency components and conformalizes high-frequency residuals via adaptive gating over a low-frequency embedding. Experimental results on real-world benchmark datasets across multiple application domains show that SCALE not only achieves valid coverage but also consistently improves the coverage-efficiency trade-off over the state-of-the-art CP methods.

## 1. Introduction

Spatio-temporal forecasting of time series is ubiquitous in critical infrastructure, ranging from intelligent transportation systems to power grid load balancing (Capone et al., 2025; Dong et al., 2025). It aims to predict the future evolution of a dynamic system observed over time and space. In practice, since the number of series can be very large and they may have complex interactions, observations are typically organized as a graph-structured multivariate time series (Jin et al., 2024; Han et al., 2024; Deng et al., 2023; Wu et al., 2020). To forecast such coupled time series, Spatio-temporal graph neural networks (STGNNs) are proposed (Capone et al., 2025). They have become the dominant approaches in time series forecasting due to their superior performance on point forecasting.

However, point forecasting alone is often insufficient in some high-stakes scenarios (Deng et al., 2025; Cao et al., 2025), where stakeholders may need uncertainty quantification (UQ) to support risk-sensitive planning. Regrettably, existing STGNNs often fall short of rigorous uncertainty quantification, rendering their predictions unreliable in some complex scenarios (Mallick et al., 2024). To derive reliable predictions, Conformal prediction (CP) has been proposed, which provides a model-agnostic framework for UQ. The common practice of CP is to transform the output of a specific forecaster into a prediction set that covers the ground-truth with high probability. To this end, conformal prediction commonly assumes *exchangeability*, which means that the joint distribution of examples in a held-out calibration set is invariant under the permutation of their indices (Angelopoulos & Bates, 2021).

Although exchangeability may hold in some static scenarios, this assumption is often violated in time series forecasting due to serial (temporal) dependence and potential distribution shift over time (Barber et al., 2023; Luo et al., 2026). To address the non-exchangeability problem, recent CP methods, such as Conformal Prediction for Time Series (CPTS) (Xu & Xie, 2023a) and Sequential Predictive Conformal Inference (SPCI) (Xu & Xie, 2023b), propose to design sequential or adaptive calibration techniques. These procedures typically rely on the identification of (approximate) exchangeability, which is further leveraged to recover

---
[*]Equal contribution [1] School of Automation and Intelligent Sensing, Shanghai Jiao Tong University, Shanghai, China. Correspondence to: Zhe Liu <liuzhesjtu@sjtu.edu.cn>, Chen Gong <chen.gong@sjtu.edu.cn>, Hesheng Wang <wanghesheng@sjtu.edu.cn>.

*Proceedings of the 43rd International Conference on Machine Learning*, Seoul, South Korea. PMLR 306, 2026. Copyright 2026 by the author(s).

prediction errors or nonconformity scores (Wu et al., 2025).

The non-exchangeability challenge becomes even more severe for *graph-structured* multivariate time series (MTS), where observations at each time step form a coupled random vector and cross-node interactions can induce system-level dependence. Such coupling can violate the exchangeability condition even when each coordinate sequence appears separately exchangeable, making a direct application of CP unreliable. In view of this point, most existing MTS forecasting approaches either apply CP independently per series (yielding only marginal guarantees) or attempt to enforce exchangeability through restrictive assumptions on inter-series relations (e.g., sparsity) (Wang et al., 2025). Nevertheless, these assumptions may not hold in practice, and learning with joint non-exchangeability still remains challenging.

In this paper, we observe that the breakdown of joint exchangeability in graph-structured multivariate time series is closely tied to strong coupling. Inspired by the theoretical works in spectral graph learning and graph signal processing (Chen et al., 2023; Isufi et al., 2024), we find that such coupling is mainly reflected by global trends, which are characterized by low-frequency components in the graph spectrum (Dabush & Routtenberg, 2024). Meanwhile, the high-frequency components are verified to have weaker cross-node interaction and are more likely to be exchangeable. This discovery is shown in Figure 1, which illustrates a clear structure of exchangeability. Therefore, to ensure effective conformal prediction, instead of directly calibrating the original non-exchangeable MTS, we can apply conformal prediction in the spectral domain.

Based on this insight, we introduce the concept of Spectral Graph Conditional Exchangeability (SGCE), which formalizes that the high-frequency components can be approximately exchangeable when conditioned on low-frequency ones. Here, the conditioning preserves the global trend in the evolution of time series, promoting a more precise prediction for each individual series. Building upon SGCE, we propose a practical implementation named Spectral Conformal prediction via wAveLEt transform (SCALE). SCALE adopts graph wavelets to decompose the original MTS into low-/high-frequency components. It then conformalizes high-frequency residual scores via adaptive gating on low-frequency components to produce prediction intervals. Compared with prior CP approaches (Cini et al., 2025) that pursue exchangeability via cross-variable encoding under the assumption of sparse dependence, our approach models joint exchangeability through spectral conditioning, providing a new paradigm for CP in graph-structured MTS.

Our main contributions are threefold:

- Algorithmically, we develop SCALE, a graph-wavelet conformal prediction framework that adapts conformal

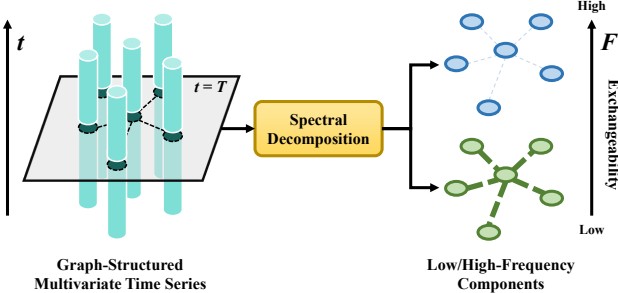

*Figure 1.* The discovery in our study. The left panel displays graph-structured MTS. At each time step, the corresponding snapshot is decomposed into multiple frequency bands (right panel), where components with higher frequencies tend to be more exchangeable due to weaker cross-node interactions.

quantiles to calibrate high-frequency components conditioned on low-frequency ones.

- Theoretically, we formalize Spectral Graph Conditional Exchangeability (SGCE) and establish coverage guarantees for conformal prediction on graph-structured MTS.

- Empirically, we demonstrate an improved coverage–efficiency trade-off on real-world benchmark datasets across diverse domains when compared with state-of-the-art baseline methods for conformal prediction in graph-structured MTS forecasting.

## 2. Related Work

In this section, we review some prior works closely related to our study, i.e., spatio-temporal graph forecasting and conformal prediction.

**Spatio-temporal graph forecasting.** Spatio-temporal graph neural networks (STGNNs) are standard paradigms for traffic and sensor-network forecasting, which combine temporal dynamics with graph-based spatial aggregation (Li et al., 2018; Yu et al., 2018; Wu et al., 2019). Prior work on STGNNs explores heterogeneous and time-varying dependencies via adaptive/dynamic adjacency, attention-based spatial mixing, and multi-component spatio-temporal blocks (Guo et al., 2019; Zheng et al., 2020; Wu et al., 2020). Benchmarking studies further document substantial heterogeneity across tasks and datasets for multivariate forecasting, motivating robust and interpretable uncertainty quantification (Shao et al., 2025). These modeling choices complicate UQ: global regime shifts can drive coherent residual changes across nodes, which makes the joint exchangeability assumption in conformal calibration hard to satisfy in multivariate time series forecasting.

**Conformal prediction.** Conformal prediction (CP) provides distribution-free uncertainty quantification by calibrating a nonconformity score and converting it into prediction sets. It has finite-sample coverage under the exchangeability assumption (Vovk et al., 2005; Shafer & Vovk, 2008).

A large body of work develops practical variants for regression and quantile-based prediction sets, including split conformal and conformalized quantile regression (Lei et al., 2018; Romano et al., 2019). These works also clarify what guarantees remain when exchangeability is violated (Barber et al., 2023; Angelopoulos & Bates, 2021). Recent studies further examine conformal selection, calibration, and robustness in deep models (Huang et al., 2024a; Xi et al., 2025a;b). In particular, for time series, due to temporal dependence and cross-series couplings, exchangeability may not hold in general. This motivates sequential, online, and adaptive conformal procedures (Xu & Xie, 2023a;b; Wu et al., 2025; Li & Rodríguez, 2025). Closely related are conformal approaches for correlated data (Cini et al., 2025) and recent work on temporal graphs (Wang et al., 2025). In our study, instead of directly applying CP to the original series, we argue that exchangeability is more likely to hold for high-frequency components in the spectral domain.

## 3. Preliminaries

This section introduces the problem formulation and the foundational concepts utilized in our framework.

### 3.1. Problem Formulation

In this paper, we focus on the problem of uncertainty quantification for multivariate time series (MTS) forecasting on graphs. We use the set $\{\mathbf{x}^1, \ldots, \mathbf{x}^N\}$ to denote a sample of $N$ correlated time series and $\mathbf{X}_{t:t+T} := [\mathbf{x}_{t:t+T}^1, \ldots, \mathbf{x}_{t:t+T}^N]^\top \in \mathbb{R}^{N \times T}$ to denote the multivariate observation. Here, $\mathbf{x}_{t:t+T}^i := [x_t^i, \ldots, x_{t+T-1}^i]^\top \in \mathbb{R}^T$ is the $i$-th series observed over the discrete time indices from $t$ to $t + T - 1$. In addition, we denote the multivariate observation at time $t$ by the snapshot $\mathbf{X}_t := [x_t^1, \ldots, x_t^N]^\top \in \mathbb{R}^{N \times 1}$. Due to severe inter-series coupling, such MTS is modeled in a graph structure $\mathcal{G} = (\mathcal{V}, \mathcal{E})$, where $\mathcal{V}$ denotes the set of $N$ nodes and $\mathcal{E}$ encodes pairwise correlations. Here, we represent $\mathcal{E}$ using an adjacency matrix $\mathbf{A} \in \mathbb{R}^{N \times N}$. Unlike approaches that treat correlations as auxiliary exogenous covariates (Cini et al., 2025), we assume that the data-generating process is conditioned on the graph topology. Specifically, each observation is generated according to a spatio-temporal conditional distribution

$$x_t^i \sim p\left(x_t^i \mid \mathbf{X}_{<t}, \mathcal{N}_{\mathcal{G}}(i)\right), \quad (1)$$

where $\mathbf{X}_{<t}$ denotes historical observations prior to time $t$, and $\mathcal{N}_{\mathcal{G}}(i)$ denotes the neighborhood of node $i$ on graph $\mathcal{G}$.

**Point forecasting.** Point forecasting aims to learn a predictor $F_{\boldsymbol{\theta}}$, parameterized by $\boldsymbol{\theta}$, that maps past observations to future values. Given a look-back window of length $W$ and a prediction horizon $K$, the predictor produces

$$\widehat{\mathbf{X}}_{t+1:t+K} = F_{\boldsymbol{\theta}}\left(\mathbf{X}_{t-W+1:t}, \mathbf{A}\right). \quad (2)$$

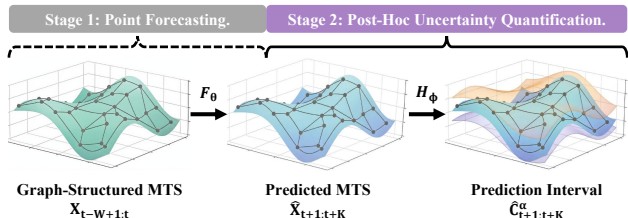

*Figure 2.* Two-stage post-hoc uncertainty quantification for graph-structured multivariate time series. In the first stage, a point forecaster $F_{\boldsymbol{\theta}}$ maps historical graph signals $\mathbf{X}_{t-W+1:t}$ to point predictions $\widehat{\mathbf{X}}_{t+1:t+K}$. In the second stage, the forecaster is fixed, and a post-hoc uncertainty quantification module $H_{\boldsymbol{\phi}}$ calibrates prediction intervals $\widehat{\mathcal{C}}_{t+1:t+K}^\alpha$ around the point forecasts. This paper focuses on the second stage.

Here, $F_{\boldsymbol{\theta}}$ can be any graph-based forecasting model, such as spatio-temporal graph neural networks (STGNNs) (Yu et al., 2018; Li et al., 2018; Wu et al., 2019). The model $F_{\boldsymbol{\theta}}$ is typically trained to minimize a deterministic loss function $\ell(\mathbf{X}_{t:t+T}, \widehat{\mathbf{X}}_{t:t+T})$, e.g., the mean squared error (MSE) loss or mean absolute error (MAE) loss (Lim & Zohren, 2021).

**Post-hoc uncertainty quantification.** Given a pre-trained predictor $F_{\boldsymbol{\theta}}$, our goal is to perform post-hoc uncertainty quantification for its point forecasts. Specifically, for a miscoverage rate $\alpha \in (0, 1)$, we aim to construct a prediction interval $\widehat{\mathcal{C}}_{t+1:t+K}^\alpha$ around the point prediction $\widehat{\mathbf{X}}_{t+1:t+K}$ produced by $F_{\boldsymbol{\theta}}$. The target coverage requirement is that the future graph signal $\mathbf{X}_{t+1:t+K}$ lies within the prediction interval with probability at least $1 - \alpha$, i.e.,

$$\mathbb{P}\left(\mathbf{X}_{t+1:t+K} \in \widehat{\mathcal{C}}_{t+1:t+K}^\alpha\right) \geq 1 - \alpha. \quad (3)$$

Figure 2 illustrates this two-stage setting. In the first stage, the graph-based forecaster $F_{\boldsymbol{\theta}}$ produces the point prediction $\widehat{\mathbf{X}}_{t+1:t+K}$ from the historical graph signal $\mathbf{X}_{t-W+1:t}$. In the second stage, the forecaster is fixed, and the post-hoc uncertainty quantification module constructs the prediction interval $\widehat{\mathcal{C}}_{t+1:t+K}^\alpha$ around this point prediction.

Conformal prediction provides a principled way to construct such prediction intervals with finite-sample marginal coverage under exchangeability (Vovk et al., 2005). As the foundation of our method, we briefly review the split conformal prediction method in the following.

### 3.2. Split Conformal Prediction

Split Conformal Prediction (SCP) (Vovk et al., 2005) uses quantiles of prediction error to construct prediction intervals because these quantiles directly characterize the tails of the error distribution. In detail, SCP adopts an auxiliary calibration set $\mathcal{T}_{\text{cal}}$ for quantile estimation, and $x_t^{i,\text{cal}}$ is denoted as the $i$-th calibration example for time step $t$. At time step $t$, SCP computes the residual error $r_t^i = x_t^{i,\text{cal}} - \widehat{x}_t^{i,\text{cal}}$, where $\widehat{x}_t^{i,\text{cal}}$ is the output of the predictor $F_{\boldsymbol{\theta}}$. These residual errors are then aggregated into residual snapshot $\mathbf{R}_t =$

$[r_t^1, \ldots, r_t^N]^\top \in \mathbb{R}^{N \times 1}$. In this way, we obtain the residual matrix $\mathbf{R}_{t-W+1:t} = [\mathbf{R}_{t-W+1}, \ldots, \mathbf{R}_t] \in \mathbb{R}^{N \times W}$. Let the $\alpha/2$- and $1-\alpha/2$-quantiles of $\mathbf{R}_{t-W+1:t}$ be $\mathbf{Q}_{t+1:t+K}^{\alpha/2}$ and $\mathbf{Q}_{t+1:t+K}^{1-\alpha/2}$. The prediction interval $\widehat{\mathcal{C}}_{t+1:t+K}^\alpha$ is given by

$$\widehat{\mathcal{C}}_{t+1:t+K}^\alpha = \left[ \widehat{\mathbf{X}}_{t+1:t+K} + \mathbf{Q}_{t+1:t+K}^{\alpha/2}, \widehat{\mathbf{X}}_{t+1:t+K} + \mathbf{Q}_{t+1:t+K}^{1-\alpha/2} \right]. \tag{4}$$

Furthermore, instead of directly using the quantiles of $\mathbf{R}_{t-W+1:t}$ to construct $\widehat{\mathcal{C}}_{t+1:t+K}^\alpha$, we can employ a network $H_\phi$ (parameterized by $\phi$) to adaptively learn such quantiles. Formally, given the residual $\mathbf{R}_{t-W+1:t}$ and the adjacency matrix $\mathbf{A}$, network $H_\phi$ outputs the estimated lower and upper quantiles $\widehat{\mathbf{Q}}_{t+1:t+K}^{\alpha/2}$ and $\widehat{\mathbf{Q}}_{t+1:t+K}^{1-\alpha/2}$, namely

$$\widehat{\mathbf{Q}}_{t+1:t+K}^{\bar{\alpha}} = H_\phi \left( \mathbf{R}_{t-W+1:t}, \mathbf{A}, \bar{\alpha} \right), \tag{5}$$

where $\bar{\alpha} \in \{\alpha/2, 1-\alpha/2\}$. The coverage guarantee of SCP fundamentally relies on the assumption that the joint distribution of data is invariant under permutation (Angelopoulos & Bates, 2021). However, this assumption is frequently violated in graph-structured MTS forecasting due to strong coupling. In view of this, we resort to spectral graph theory to address this limitation, and the critical techniques used in our method are introduced in the next section.

### 3.3. Spectral Graph Wavelet Transform

In this section, we introduce the Spectral Graph Wavelet Transform (SGWT) (Hammond et al., 2011; Shuman et al., 2013), which will be used later to perform spectral decomposition. SGWT is designed to filter graph signals in the spectral domain. Such filtering relies on the normalized Laplacian matrix $\mathbf{\Delta} = \mathbf{I} - \mathbf{D}^{-\frac{1}{2}} \mathbf{A} \mathbf{D}^{-\frac{1}{2}}$, where $\mathbf{A}$ is the adjacency matrix and $\mathbf{D}$ is the degree matrix. In spectral graph theory, the eigendecomposition of $\mathbf{\Delta}$ is defined as $\mathbf{\Delta} = \mathbf{U}\mathbf{\Lambda}\mathbf{U}^\top$, where the columns of $\mathbf{U}$ are the orthonormal eigenvectors and $\mathbf{\Lambda}$ is the diagonal matrix of eigenvalues. These eigenvalues are typically interpreted as frequencies in the graph spectral domain. In SGWT, by employing a set of band-pass kernels $\{g_s\}_{s=1}^S$ and a low-pass kernel $h$, the wavelet coefficients for a signal snapshot $\mathbf{X}_t \in \mathbb{R}^{N \times 1}$ at scale $s$ can be computed via spectral convolution

$$\mathbf{W}_{s,t} = \mathbf{U} g_s(\mathbf{\Lambda}) \mathbf{U}^\top \mathbf{X}_t, \tag{6}$$

where the diagonal matrix $g_s(\mathbf{\Lambda})$ acts as a band-pass filter. Moreover, the low-pass kernel $h(\mathbf{\Lambda})$ extracts the global trends, yielding $\mathbf{V}_t = \mathbf{U} h(\mathbf{\Lambda}) \mathbf{U}^\top \mathbf{X}_t$. Formally, let $\mathbf{\Phi}$ denote the spectral operator representing this multi-scale transformation, and then the input signal $\mathbf{X}_t$ can be decomposed into global trends and multi-scale details, namely

$$\mathbf{\Phi} \mathbf{X}_t = \mathbf{V}_t + \sum_{s=1}^S \mathbf{W}_{s,t}. \tag{7}$$

Based on Eq. (7), the estimated low-frequency and high-frequency components are typically denoted as $\widehat{\mathbf{L}}_t$ and $\widehat{\mathbf{H}}_t$, respectively. Given a cutoff scale $k$ (with $1 \leq k \leq S$), the estimation $\widehat{\mathbf{L}}_t$ is constructed by combining the low-pass output with wavelet coefficients at scales $s \geq k$, i.e., $\widehat{\mathbf{L}}_t = \mathbf{V}_t + \sum_{s=k}^S \mathbf{W}_{s,t}$. The high-frequency component $\widehat{\mathbf{H}}_t$ is obtained from the remaining wavelet coefficients at scales $s < k$, namely $\widehat{\mathbf{H}}_t = \sum_{s=1}^{k-1} \mathbf{W}_{s,t}$.

## 4. Methodology

In this section, we present the motivation for our study and describe the proposed SCALE method in detail.

### 4.1. Motivation: Exchangeability in Spectral Domain

In the previous section, we have presented the general procedure to derive prediction intervals via SCP. The success of SCP mainly relies on the assumption that the data distribution remains fixed under permutation. Formally, for a calibration set of variables $\{\mathbf{x}_1, \ldots, \mathbf{x}_n\}$ and any permutation $\sigma$, the joint distribution must satisfy

$$p(\mathbf{x}_1, \ldots, \mathbf{x}_n) = p(\mathbf{x}_{\sigma(1)}, \ldots, \mathbf{x}_{\sigma(n)}). \tag{8}$$

This strict exchangeability in data distribution also implies that the joint distribution of the variables remains invariant regardless of the ordering, which makes SCP satisfy the coverage guarantee in Eq. (3). However, such an assumption is often infeasible for graph-structured MTS, which have complex spatial couplings and temporal dependencies. Even more seriously, *node-wise exchangeability* of individual time series $\mathbf{x}_t^i$ does not imply the *system-wide exchangeability* of the snapshots $\mathbf{X}_t$. Formally, even if every time series satisfies exchangeability, the sequence of system-wide snapshots may not, namely

$$p(\mathbf{x}_1^i, \ldots, \mathbf{x}_n^i) = p(\mathbf{x}_{\sigma(1)}^i, \ldots, \mathbf{x}_{\sigma(n)}^i), \ \forall i \in \{1, \cdots, N\}$$
$$\nRightarrow \quad p(\mathbf{X}_1, \ldots, \mathbf{X}_n) = p(\mathbf{X}_{\sigma(1)}, \ldots, \mathbf{X}_{\sigma(n)}). \tag{9}$$

We quantify this system-level violation by defining the permutation-based exchangeability gap:

$$\text{ExGap}(\mathbf{X}) = \mathbb{E}_\sigma \left[ \text{MMD} \left( \mathcal{B}_w(\mathbf{X}), \mathcal{B}_w(\mathbf{X}_\sigma) \right) \right], \tag{10}$$

where $\mathbb{E}_\sigma$ averages over random temporal permutations, $\mathcal{B}_w(\mathbf{X})$ denotes the collection of vectorized system-wide temporal blocks with length $w$, $\mathbf{X}_\sigma$ is the temporally permuted sequence, and MMD denotes the maximum mean discrepancy (Gretton et al., 2012). A smaller $\text{ExGap}(\mathbf{X})$ indicates weaker sensitivity to temporal permutation, i.e., a higher empirical degree of exchangeability. The detailed construction is provided in Appendix A.

Accordingly, directly applying the standard conformal prediction method, such as SCP, to the forecasting of graph-structured MTS can lead to unreliable prediction intervals.

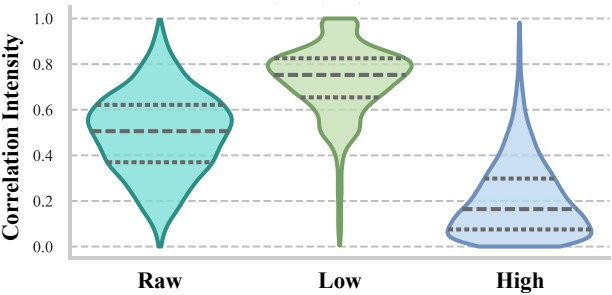

*Figure 3.* Distributions of the correlation intensity for the original MTS, the low-frequency components, and the high-frequency components on the METR-LA dataset.

Inspired by the theoretical works in spectral graph learning and graph signal processing (Hammond et al., 2011; Shen et al., 2021), we find that although exchangeability can hardly hold for original graph-structured MTS, it can be easily satisfied in the spectral domain. Specifically, the spectral graph theory demonstrates that the main factors for non-exchangeability, namely the cross-node coupling and global trends, are essentially characterized as low-frequency components in the graph spectrum. Meanwhile, the high-frequency components typically encode local variations. Such local variations are more nearly exchangeable when compared with the low-frequency ones.

To justify this point, we perform the correlation analysis on MTS from the METR-LA dataset (Li et al., 2018). For time series $\{\mathbf{x}^1, \ldots, \mathbf{x}^N\}$, we compute the Pearson correlation coefficients $\rho_{i,j}$ for $\mathbf{x}^i$ and $\mathbf{x}^j$, $\forall\, i, j \in \{1, \cdots, N\}$. For each time series $\mathbf{x}^i$, we further define its *correlation intensity* $c_i$ as the average correlation with other series, namely $c_i = \frac{\sum_{j \neq i} \rho_{i,j}}{N-1}$. The calculation of intensity $c_i$ for low- and high-frequency components is analogous. As shown in Figure 3, although the low- and high-frequency components exhibit comparable energy levels, the high-frequency components show weaker correlation intensities, indicating reduced cross-node couplings. This reduction in coupling provides empirical support for stronger exchangeability. Across datasets from diverse domains, we examine the relationship between the dataset-level coupling strength $\bar{c}(\mathbf{X}) = \frac{1}{N} \sum_{i=1}^{N} c_i(\mathbf{X})$ and the permutation-based exchangeability gap $\text{ExGap}(\mathbf{X})$. We observe a strong positive correlation between $\bar{c}(\mathbf{X})$ and $\text{ExGap}(\mathbf{X})$, with a Pearson correlation of **0.885** and a Spearman correlation of **0.927**. This indicates that weaker cross-node coupling is consistently associated with smaller deviations from exchangeability. Consequently, by applying spectral decomposition to extract high-frequency components with reduced coupling, we obtain representations that are closer to exchangeability.

Based on the above observations, we can decompose the original MTS in the spectral domain to ensure exchangeability for conformal prediction. However, only using exchangeable high-frequency components for prediction may not be

sufficient, as low-frequency components are also essential in capturing global trend information. Such information serves as guidance to promote accurate prediction for each individual series. Therefore, the low-frequency components are considered as conditions while the high-frequency components are adopted for reliable conformal calibration.

As introduced in Section 3.2, we denote the low- and high-frequency components of residual $\mathbf{R}_t$ as $\mathbf{L}_t$ and $\mathbf{H}_t$, respectively. Moreover, we also denote the windowed low- and high-frequency components by $\mathcal{L}_t \triangleq \mathbf{L}_{t-W+1:t}$ and $\mathcal{H}_t \triangleq \mathbf{H}_{t-W+1:t}$, respectively. Based on these definitions, we propose an important concept termed *Spectral Graph Conditional Exchangeability* in Definition 4.1.

**Definition 4.1** (Spectral Graph Conditional Exchangeability, SGCE)**.** The residual process satisfies *Spectral Graph Conditional Exchangeability* if the high-frequency sequences $\{\mathcal{H}_t\}_t$ are exchangeable conditional on respective components $\{\mathcal{L}_t\}_t$. Formally, for any sequence of time indices $\{t_1, \ldots, t_n\}$ and any permutation $\sigma$, it holds that

$$
\begin{aligned}
&p(\mathcal{H}_{t_1}, \ldots, \mathcal{H}_{t_n} \mid \mathcal{L}_{t_1}, \ldots, \mathcal{L}_{t_n}) \\
=\ &p(\mathcal{H}_{t_{\sigma(1)}}, \ldots, \mathcal{H}_{t_{\sigma(n)}} \mid \mathcal{L}_{t_{\sigma(1)}}, \ldots, \mathcal{L}_{t_{\sigma(n)}}).
\end{aligned}
\tag{11}
$$

This definition is subsequently adopted to design practical algorithms for reliable conformal prediction.

### 4.2. Implementation: Spectral Conformal Prediction via Wavelet Transform

Based on the concept of SGCE in Definition 4.1, we further provide a practical implementation termed Spectral Conformal prediction via wAveLEt transform (SCALE). The overall framework is illustrated in Figure 4. In detail, we apply SGWT to the residuals and derive the low/high-frequency components $\widehat{\mathbf{L}}_t$ and $\widehat{\mathbf{H}}_t$. The calculation of them is provided in Section 3.3. Moreover, the cutoff scale can be auto-selected by a lightweight SGWT diagnostic, and the pseudocode is provided in Appendix D. The frequency components subsequently serve as inputs in our method.

Following Definition 4.1, we operationalize SGCE in the wavelet domain with two lightweight pathways: a learnable low-frequency encoder that maps $\widehat{\mathbf{L}}_{t-W+1:t}$ to the conditioning embedding $\mathbf{C}_t$, and a parameter-free high-frequency statistics extractor that maps $\widehat{\mathbf{H}}_{t-W+1:t}$ to the feature summary $\mathbf{M}_t$ (Liu et al., 2025).

**Processing of low/high-frequency components.** The decomposition above yields two streams with distinct roles. On the low-frequency side, we adopt an STID-style encoder to obtain the conditioning embedding (Shao et al., 2022a):

$$
\mathbf{C}_t = \text{MLP}\Big( E_x(\widehat{\mathbf{L}}_{t-W+1:t}) \,\|\, E_s(\mathbf{v}) \,\|\, E_p(t) \Big), \tag{12}
$$

where $\|$ denotes concatenation, $\widehat{\mathbf{L}}_{t-W+1:t}$ is the low-frequency history, $\mathbf{v}$ is node identity, and $E_x$, $E_s$, $E_p$ are

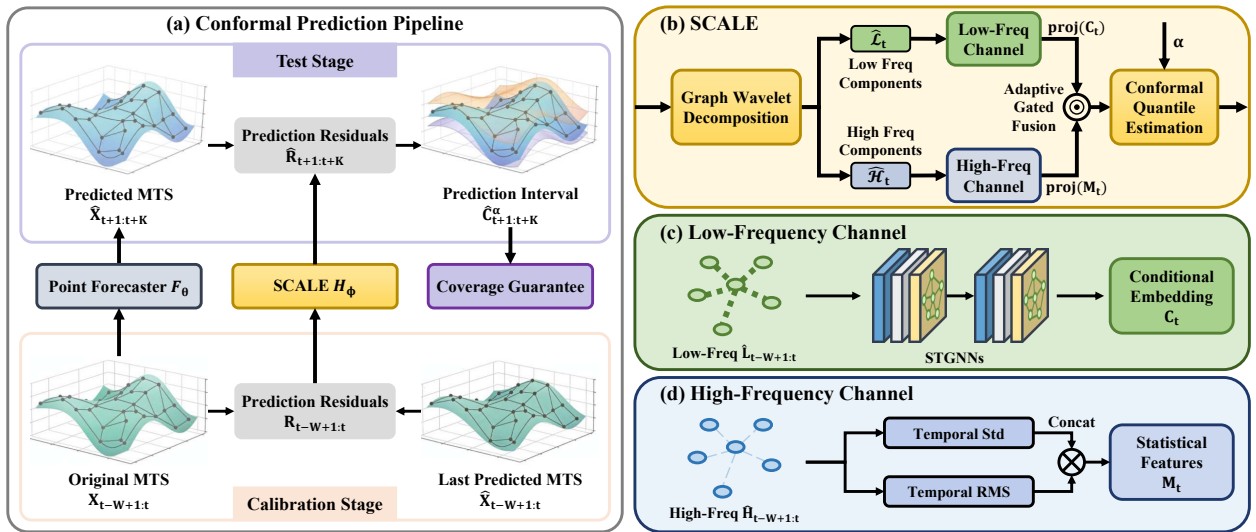

*Figure 4.* **The framework of Spectral Conformal prediction via wAveLEt transform (SCALE). (a) Workflow:** SCALE integrates into the standard split conformal prediction pipeline, serving as an estimator for the prediction intervals. **(b) Spectral Decoupling:** Input residuals are decomposed via SGWT into low- and high-frequency components to reduce non-exchangeability while encoding global trends. **(c) Low-Frequency Channel:** A lightweight learnable encoder maps the low-frequency trends into a structural conditioning embedding $\mathbf{C}_t$. **(d) High-Frequency Channel:** A parameter-free extractor computes temporal statistics $\mathbf{M}_t$, avoiding the reintroduction of spatial couplings. Finally, the two representations are fused via an adaptive gated mechanism to output the calibrated prediction interval.

the low-frequency, spatial, and periodic-time embeddings, respectively (with $E_p$ concatenating time-of-day and day-of-week) (Zheng et al., 2020; Huang et al., 2024b).

On the high-frequency side, $\widehat{\mathbf{H}}_t$ mainly captures localized transient fluctuations with weak cross-node dependence as illustrated in Figure 3. Therefore, introducing an additional learnable spatio-temporal backbone on $\widehat{\mathbf{H}}_t$ is unnecessary and may even reintroduce spatial coupling into the residual representation, degrading calibration robustness. Moreover, unlike standard representation learning tasks that require highly expressive latent features, conformal calibration mainly depends on estimating local uncertainty and fluctuation scales of residuals. Prior conformal prediction studies have shown that reliable interval calibration can already be achieved through relatively simple score- or quantile-based statistics without requiring a highly expressive model (Lei et al., 2018; Romano et al., 2019). Accordingly, we summarize $\widehat{\mathbf{H}}_{t-W+1:t}$ using lightweight per-node temporal statistics as the high-frequency representation for subsequent conditional quantile modulation:

$$\text{STD}(\widehat{\mathbf{H}}_{t-W+1:t}) = \sqrt{\frac{1}{W}\sum_{\tau=t-W+1}^{t}(\widehat{\mathbf{H}}_\tau - \bar{\mathbf{H}})^2}, \quad (13a)$$

$$\text{RMS}(\widehat{\mathbf{H}}_{t-W+1:t}) = \sqrt{\frac{1}{W}\sum_{\tau=t-W+1}^{t}\widehat{\mathbf{H}}_\tau^2}, \quad (13b)$$

$$\mathbf{M}_t = \text{STD}(\widehat{\mathbf{H}}_{t-W+1:t}) \,\|\, \text{RMS}(\widehat{\mathbf{H}}_{t-W+1:t}), \quad (13c)$$

where $\bar{\mathbf{H}} = \frac{1}{W}\sum_{\tau=t-W+1}^{t}\widehat{\mathbf{H}}_\tau$ is the temporal mean. Here,

STD measures the fluctuation scale of the high-frequency residuals within the look-back window, while RMS characterizes their overall magnitude and energy level through square-and-average aggregation. Together, these two statistics summarize the aspects of high-frequency components most directly related to interval adaptation, namely local variability and residual intensity, while avoiding the introduction of unnecessary learnable spatial couplings.

Building upon these representations, SCALE introduces a *low-frequency conditioned* quantile predictor. As shown in Figure 4, the low-frequency branch outputs an embedding $\mathbf{C}_t$, which is then projected to a quantile-channel feature map $\mathbf{Z}_{t+1:t+K}^{\mathrm{L}}$. Moreover, a high-frequency branch maps $\mathbf{M}_t$ to a parallel feature map $\mathbf{Z}_{t+1:t+K}^{\mathrm{H}}$. Conditioning is implemented via an adaptive gate where the low-frequency embedding generates a gating map (Xiong et al., 2024):

$$\mathbf{G}_{t+1:t+K} = \text{SIGMOID}\big(\text{PROJ}_{\text{gate}}(\mathbf{C}_t)\big), \quad (14a)$$

$$\widehat{\mathbf{Q}}_{t+1:t+K}^{\bar{\alpha}} = \mathbf{Z}_{t+1:t+K}^{\mathrm{L}} + \mathbf{G}_{t+1:t+K} \odot \mathbf{Z}_{t+1:t+K}^{\mathrm{H}}. \quad (14b)$$

Here, $\widehat{\mathbf{Q}}_{t+1:t+K}^{\bar{\alpha}}$ denotes the predicted residual quantile sequence conditioned on $(\mathbf{M}_t; \mathbf{C}_t)$, where $\bar{\alpha} \in \mathcal{T}_\alpha$ and $\mathcal{T}_\alpha = \{\alpha/2, 1-\alpha/2\}$. Thus, SCALE estimates both the lower and upper residual quantiles under the same spectral conditioning information, which are then used to construct two-sided prediction intervals. During training, $r$ denotes an element of the future residual target $\mathbf{R}_{t+1:t+K} = \mathbf{X}_{t+1:t+K} - \widehat{\mathbf{X}}_{t+1:t+K}$, and the loss is applied to residual entries across nodes. The quantile predictor is trained end-to-end with the pinball loss, i.e., the standard

quantile regression loss (Koenker & Bassett Jr., 1978):

$$
\ell_\tau(r, \widehat{q}) = \begin{cases} \tau(r - \widehat{q}), & r \geq \widehat{q}, \\ (\tau - 1)(r - \widehat{q}), & r < \widehat{q}, \end{cases} \quad \tau \in \mathcal{T}_\alpha. \quad (15)
$$

## 5. Theoretical Analyses

In this section, we provide theoretical analyses for the proposed SCALE method, including finite-sample coverage guarantees for prediction intervals under perfect and imperfect approximation of wavelet transform.

Based on SGCE (Definition 4.1), we derive the following validity guarantee for the proposed SCALE method.

**Theorem 5.1** (Finite-Sample Coverage under SGCE). *Suppose that the frequency components of the residual* $\mathbf{R}_{t-W+1:t}$ *satisfy SGCE (Definition 4.1). For* $\forall\, \alpha \in (0, 1)$, *let* $\widehat{\mathcal{C}}_{t+1:t+K}^{X,\alpha}$ *denote the prediction interval of our SCALE method, and then it holds that*

$$
\mathbb{P}\left(\mathbf{X}_{t+1:t+K} \in \widehat{\mathcal{C}}_{t+1:t+K}^{X,\alpha}\right) \geq 1 - \alpha. \quad (16)
$$

Theorem 5.1 guarantees that our framework maintains finite-sample validity regardless of the distribution shifts in the low-frequency trends $\mathcal{L}_t$, provided that the high-frequency components $\mathcal{H}_t$ remain conditionally exchangeable. The formal statement and proof are provided in Appendix B.

**Robustness to wavelet approximation.** In practice, the components $\widehat{\mathbf{L}}_t$ and $\widehat{\mathbf{H}}_t$ obtained via SGWT are *empirical estimates*. To bridge the gap between ideal spectral decoupling and practical implementation, we provide a robust validity guarantee that explicitly accounts for wavelet approximation error and high-frequency couplings.

**Theorem 5.2** (Informal: Approximate Coverage under Imperfect Spectral Decomposition). *Suppose that the spectral decomposition admits bounded reconstruction error and high-frequency components exhibit bounded deviation from exchangeability. For* $\forall \alpha \in (0, 1)$, *let* $\widehat{\mathcal{C}}_{t+1:t+K}^{sgwt,\alpha}$ *be the prediction interval constructed via graph wavelet transform and spectral-domain conformal calibration. Then,* $\widehat{\mathcal{C}}_{t+1:t+K}^{sgwt,\alpha}$ *satisfies the approximate coverage guarantee*

$$
\mathbb{P}\left(\mathbf{X}_{t+1:t+K} \in \widehat{\mathcal{C}}_{t+1:t+K}^{sgwt,\alpha}\right) \geq 1 - \alpha - \delta, \quad (17)
$$

*where the coverage gap* $\delta$ *decomposes as* $\delta = \delta_{leak} + \delta_{dep}$. *Here,* $\delta_{leak}$ *and* $\delta_{dep}$ *capture spectral leakage effects and spatial couplings, respectively.*

**Remark.** The formal statement of Theorem 5.2 is provided in Appendix C, which formalizes the trade-off between implementation constraints and theoretical validity. It demonstrates that the coverage gap is controllable: as the spectral filter approaches an ideal band-pass ($\delta_{leak} \to 0$) and the

high-frequency couplings vanish ($\delta_{dep} \to 0$), the total gap disappears ($\delta \to 0$). This implies that our framework *asymptotically recovers* the finite-sample coverage guarantee.

## 6. Experimental Results

In this section, we show experimental results on datasets from multiple domains to show the effectiveness of our SCALE and the validity of the prediction intervals. The official implementation is available at `https://github.com/IRMVLab/SCALE.git`.

### 6.1. Experimental Settings

**Experimental Setup.** Our experimental pipeline consists of two stages. In the first stage, residual sequences are obtained as the difference between the predictions of a backbone forecaster and the ground truth. In the second stage, prediction intervals are evaluated post hoc on these residuals. We adopt the common 40%/40%/20% splits for training, calibration, and testing, respectively. All baseline methods operate on the same residual tensors produced by an identical backbone run, with all other settings held fixed for fairness. In the first stage, we adopt STGNN backbone (Cini et al., 2025) for alignment, and we report results with the alternative backbones in Appendix F.2 to demonstrate the generality of our method. Across datasets, we keep the backbone forecasting settings unchanged, e.g., $W = 12$ and $K = 1$. For SGWT, we fix the kernel family and use the common scale count $S = 4$ and select the cutoff $k$ following Appendix D.

**Baseline methods.** We compare SCALE against the following baselines: 1) **SCP** (Vovk et al., 2005), the standard split conformal predictor using empirical residual quantiles; 2) **SeqCP** (Xu & Xie, 2023b), which computes empirical quantiles using only the most recent $M_{seq}$ residuals at each step; 3) **NexCP** (Barber et al., 2023), which assigns exponentially decaying weights to past residuals; 4) **EnbPI** (Xu & Xie, 2023a), which builds prediction intervals from bootstrap ensembles and an online residual buffer; 5) **HopCPT** (Auer et al., 2023), which reweights past residuals using a Hopfield-style memory; and 6) **CoREL** (Cini et al., 2025), which models cross-series dependence via a relational quantile predictor. For the multi-step analysis, we additionally include **ConForME** (Galvão Lopes et al., 2024), which extends split conformal forecasting to the multi-horizon setting by exploiting temporal dependence to calibrate efficient prediction intervals with probabilistic joint coverage.

**Datasets.** We evaluate on four widely used graph-structured multivariate time series benchmarks spanning multiple domains: **METR-LA** (Li et al., 2018) from the traffic domain, **AirQuality** (Zheng et al., 2015) from the environmental domain, **Electricity** (Trindade, 2015) from the energy domain,

*Table 1.* Residual-interval results across multiple domains on METR-LA (traffic), AirQuality (environment), Electricity (energy), and USHCN-West (climate) at $\alpha \in \{0.05, 0.1, 0.2\}$. Coverage, PI-Width, and Winkler are reported as mean±std across seeds for trainable methods; SCP/SeqCP/NexCP are single-run because they are deterministic given residuals.

| Method | METR-LA | | | AirQuality | | | Electricity | | | USHCN-West | | |
|---|---|---|---|---|---|---|---|---|---|---|---|---|
| | Coverage ↑ | PI-Width ↓ | Winkler ↓ | Coverage ↑ | PI-Width ↓ | Winkler ↓ | Coverage ↑ | PI-Width ↓ | Winkler ↓ | Coverage ↑ | PI-Width ↓ | Winkler ↓ |
| $\alpha = 0.05$ | | | | | | | | | | | | |
| SCP (Vovk et al., 2005) | 0.9470±0.0000✓ | 13.62±0.00 | 21.98±0.00 | 0.9645±0.0000✓ | 80.27±0.00 | 116.54±0.00 | 0.9488±0.0000✓ | 1043.78±0.00 | 1515.91±0.00 | 0.9477±0.0000✓ | 15.38±0.00 | 20.40±0.00 |
| SeqCP (Xu & Xie, 2023b) | 0.9097±0.0000✗ | 12.02±0.00 | 22.24±0.00 | 0.9290±0.0000✗ | 54.20±0.00 | 101.02±0.00 | 0.9310±0.0000✗ | 988.57±0.00 | 1413.55±0.00 | 0.9279±0.0000✗ | 14.23±0.00 | 20.22±0.00 |
| CoREL (Cini et al., 2025) | 0.9509±0.0041✓ | 12.19±0.44 | 18.10±0.24 | 0.9536±0.0212✓ | 59.77±3.37 | 89.33±2.08 | 0.9459±0.0279✓ | 1005.46±108.11 | 1559.68±68.70 | 0.9478±0.0024✓ | 14.59±0.14 | 18.95±0.04 |
| NexCP (Barber et al., 2023) | 0.9433±0.0000✓ | 13.33±0.00 | 21.20±0.00 | 0.9489±0.0000✓ | 58.71±0.00 | 97.72±0.00 | 0.9438±0.0000✓ | 1042.82±0.00 | 1403.47±0.00 | 0.9449±0.0000✓ | 15.19±0.00 | 19.77±0.00 |
| EnbPI (Xu & Xie, 2023a) | 0.9488±0.0000✓ | 13.65±0.00 | 21.90±0.00 | 0.9664±0.0000✓ | 74.83±0.02 | 109.94±0.04 | 0.9427±0.0000✓ | 965.59±0.24 | 1438.14±0.32 | 0.9473±0.0000✓ | 15.16±0.00 | 20.11±0.00 |
| HopCPT (Auer et al., 2023) | 0.9349±0.0025✓ | 13.16±0.35 | 21.32±0.60 | 0.9523±0.0047✓ | 59.08±3.77 | 94.43±3.90 | 0.9307±0.0121✓ | 1050.82±467.32 | 1389.08±480.94 | 0.9415±0.0010✓ | 18.48±0.17 | 24.05±0.27 |
| ConForME (Galvão Lopes et al., 2024) | 0.9471±0.0000✓ | 13.84±0.00 | 22.61±0.00 | 0.9692±0.0000✓ | 84.07±0.00 | 119.26±0.00 | 0.9507±0.0000✓ | 1064.46±0.00 | 1551.03±0.00 | 0.9485±0.0000✓ | 15.48±0.00 | 20.51±0.00 |
| SCALE w/o LF | 0.9473±0.0010✓ | 13.22±0.07 | 20.14±0.01 | 0.1688±0.0551✗ | 2.49±0.88 | 331.45±13.54 | 0.3946±0.0490✗ | 64.29±9.85 | 6106.93±113.78 | 0.8255±0.0192✗ | 9.75±0.50 | 26.79±1.23 |
| SCALE w/o SGWT | 0.9448±0.0061✓ | 11.99±0.43 | 18.76±0.06 | 0.9618±0.0076✓ | 60.83±4.87 | 88.79±1.26 | 0.9580±0.0115✓ | 918.76±76.90 | 1385.93±13.41 | 0.9454±0.0045✓ | 14.68±0.36 | 19.39±0.02 |
| SCALE | 0.9436±0.0030✓ | **11.69±0.23** | **17.92±0.07** | 0.9572±0.0046✓ | **55.38±1.94** | **84.52±0.33** | 0.9467±0.0076✓ | **776.05±38.58** | **1321.51±17.56** | 0.9444±0.0043✓ | **14.22±0.31** | **18.80±0.03** |
| $\alpha = 0.10$ | | | | | | | | | | | | |
| SCP (Vovk et al., 2005) | 0.8950±0.0000✓ | 9.46±0.00 | 16.79±0.00 | 0.9258±0.0000✗ | 56.08±0.00 | 83.39±0.00 | 0.8998±0.0000✓ | 747.06±0.00 | 1207.37±0.00 | 0.8965±0.0000✓ | 12.27±0.00 | 17.07±0.00 |
| SeqCP (Xu & Xie, 2023b) | 0.8554±0.0000✗ | 8.79±0.00 | 16.65±0.00 | 0.8791±0.0000✗ | 38.38±0.00 | 73.10±0.00 | 0.8807±0.0000✓ | 746.49±0.00 | 1160.62±0.00 | 0.8774±0.0000✗ | 11.70±0.00 | 16.91±0.00 |
| CoREL (Cini et al., 2025) | 0.9031±0.0084✓ | 9.15±0.34 | 14.30±0.14 | 0.9223±0.0107✗ | 43.32±1.29 | 66.63±0.56 | 0.8918±0.0538✓ | 748.15±90.02 | 1206.66±49.53 | 0.8979±0.0030✓ | 11.86±0.11 | 16.04±0.03 |
| NexCP (Barber et al., 2023) | 0.8934±0.0000✓ | 9.33±0.00 | 16.07±0.00 | 0.9020±0.0000✓ | **40.36±0.00** | 71.41±0.00 | 0.8939±0.0000✓ | 777.85±0.00 | 1160.19±0.00 | 0.8952±0.0000✓ | 12.22±0.00 | 16.69±0.00 |
| EnbPI (Xu & Xie, 2023a) | 0.8986±0.0000✓ | 9.45±0.00 | 16.67±0.00 | 0.9288±0.0000✗ | 51.68±0.01 | 78.73±0.02 | 0.8901±0.0000✓ | 701.52±0.17 | 1156.80±0.21 | 0.8966±0.0000✓ | 12.14±0.00 | 16.88±0.00 |
| HopCPT (Auer et al., 2023) | 0.8832±0.0040✓ | 9.84±0.13 | 16.52±0.40 | 0.9067±0.0082✓ | 42.47±1.88 | 70.28±1.85 | 0.8916±0.0152✓ | 873.83±386.28 | 1140.75±419.67 | 0.8932±0.0009✓ | 15.08±0.16 | 20.37±0.24 |
| ConForME (Galvão Lopes et al., 2024) | 0.8955±0.0000✓ | 9.34±0.00 | 17.09±0.00 | 0.9372±0.0000✓ | 58.32±0.00 | 84.45±0.00 | 0.9049±0.0000✓ | 770.72±0.00 | 1238.30±0.00 | 0.8976±0.0000✓ | 12.32±0.00 | 17.14±0.00 |
| SCALE w/o LF | 0.8968±0.0019✓ | 9.56±0.07 | 15.72±0.00 | 0.1015±0.0337✗ | 1.44±0.52 | 174.62±3.71 | 0.3212±0.0495✗ | 50.30±9.09 | 3162.84±51.73 | 0.7324±0.0158✗ | 7.81±0.37 | 20.89±0.55 |
| SCALE w/o SGWT | 0.8939±0.0106✓ | **8.88±0.37** | 14.66±0.02 | 0.9205±0.0151✗ | 44.84±3.95 | 67.44±0.87 | 0.9064±0.0218✓ | 694.12±57.09 | 1098.45±6.27 | 0.8943±0.0073✓ | 11.91±0.31 | 16.38±0.01 |
| SCALE | 0.8913±0.0039✓ | 8.89±0.16 | **14.27±0.03** | 0.9126±0.0087✓ | 41.22±1.46 | **64.54±0.23** | 0.8915±0.0127✓ | **599.07±29.49** | **1040.56±10.74** | 0.8924±0.0068✓ | **11.60±0.25** | **15.93±0.02** |
| $\alpha = 0.20$ | | | | | | | | | | | | |
| SCP (Vovk et al., 2005) | 0.7933±0.0000✓ | 5.86±0.00 | 12.18±0.00 | 0.8394±0.0000✗ | 35.70±0.00 | 57.77±0.00 | 0.8009±0.0000✓ | 501.19±0.00 | 917.56±0.00 | 0.7951±0.0000✓ | 9.08±0.00 | 13.83±0.00 |
| SeqCP (Xu & Xie, 2023b) | 0.7573±0.0000✗ | 5.87±0.00 | 11.99±0.00 | 0.7839±0.0000✗ | **25.48±0.00** | 51.77±0.00 | 0.7827±0.0000✓ | 510.49±0.00 | 895.30±0.00 | 0.7801±0.0000✓ | 8.86±0.00 | 13.71±0.00 |
| CoREL (Cini et al., 2025) | 0.8061±0.0054✓ | 6.57±0.11 | **10.96±0.06** | 0.8154±0.0503✓ | 30.19±1.00 | 49.91±0.50 | 0.7891±0.0765✓ | 521.02±44.56 | 906.88±27.66 | 0.7971±0.0053✓ | 8.96±0.10 | 13.18±0.02 |
| NexCP (Barber et al., 2023) | 0.7930±0.0000✓ | 5.95±0.00 | 11.67±0.00 | 0.8054±0.0000✓ | 26.17±0.00 | 50.99±0.00 | 0.7950±0.0000✓ | 521.63±0.00 | 894.52±0.00 | 0.7961±0.0000✓ | 9.10±0.00 | 13.61±0.00 |
| EnbPI (Xu & Xie, 2023a) | 0.7986±0.0000✓ | 5.87±0.00 | 12.05±0.00 | 0.8442±0.0001✗ | 32.62±0.01 | 54.68±0.01 | 0.7867±0.0000✓ | 478.77±0.13 | 885.06±0.16 | 0.7952±0.0000✓ | 9.00±0.00 | 13.70±0.00 |
| HopCPT (Auer et al., 2023) | 0.7796±0.0067✗ | 6.88±0.05 | 12.46±0.25 | 0.8126±0.0125✓ | 28.68±0.80 | 51.38±0.59 | 0.8074±0.0200✓ | 672.68±301.54 | 921.28±351.66 | 0.7953±0.0010✓ | 11.35±0.14 | 16.71±0.20 |
| ConForME (Galvão Lopes et al., 2024) | 0.7939±0.0000✓ | **5.80±0.00** | 12.28±0.00 | 0.8632±0.0000✗ | 36.55±0.00 | 57.72±0.00 | 0.8151±0.0000✓ | 526.62±0.00 | 945.94±0.00 | 0.7956±0.0000✓ | 9.13±0.00 | 13.89±0.00 |
| SCALE w/o LF | 0.7973±0.0031✓ | 6.54±0.06 | 11.83±0.00 | 0.0749±0.0097✗ | 1.04±0.13 | 89.32±0.49 | 0.2776±0.0261✗ | 42.38±4.21 | 1625.41±12.12 | 0.6588±0.0373✗ | 6.51±0.60 | 14.89±0.39 |
| SCALE w/o SGWT | 0.7928±0.0173✓ | 6.30±0.29 | 11.15±0.01 | 0.8311±0.0246✗ | 30.62±2.71 | 50.12±0.51 | 0.8045±0.0321✓ | 494.81±39.81 | 843.83±3.13 | 0.7925±0.0095✓ | 8.96±0.22 | 13.41±0.01 |
| SCALE | 0.7877±0.0045✓ | 6.37±0.13 | 10.99±0.02 | 0.8198±0.0136✓ | 28.89±1.02 | **48.47±0.15** | 0.7831±0.0187✓ | **435.88±21.32** | **799.91±6.42** | 0.7907±0.0089✓ | **8.79±0.19** | **13.10±0.01** |

and **USHCN-West** (Menne et al., 2009) from the climate domain. These datasets cover diverse graph sizes, sampling rates, and dynamics, providing a reasonable test bed for assessing both coverage validity and sharpness across heterogeneous settings.

**Evaluation Metrics.** We report three metrics at $\alpha \in \{0.05, 0.1, 0.2\}$. Given prediction interval $[\hat{L}_i, \hat{U}_i]$ and ground truth $y_i$, **Coverage** is the fraction of entries satisfying $y_i \in [\hat{L}_i, \hat{U}_i]$ and should be close to $1 - \alpha$. **PI-Width** measures interval sharpness by the average length $\hat{U}_i - \hat{L}_i$, where lower is better under comparable coverage (Gneiting & Raftery, 2007). **Winkler** further penalizes missed coverage, with per-entry score $(\hat{U}_i - \hat{L}_i) + \frac{2}{\alpha}(\hat{L}_i - y_i)\mathbf{1}\{y_i < \hat{L}_i\} + \frac{2}{\alpha}(y_i - \hat{U}_i)\mathbf{1}\{y_i > \hat{U}_i\}$; lower is better (Winkler, 1972). Coverage entries include a check or cross depending on whether the absolute coverage error is within 0.02.

### 6.2. Comparison Experimental Results

Table 1 shows that SCALE maintains near-nominal coverage, empirically supporting the finite-sample guarantee in Theorem 5.1. Simultaneously, it achieves the strongest efficiency; notably, on Electricity ($\alpha = 0.05$), SCALE reduces the PI width by $\approx 19.6\%$ over EnbPI, the strongest competing baseline under this setting. This gain validates our motivation (Section 4.1) that spectral decomposition effectively decouples spatial correlations, enabling tighter calibration on high-frequency components compared to conservative baselines. Furthermore, the superiority over CoREL highlights the advantage of our SGCE framework: explic-

itly conditioning on low-frequency trends ensures sharper intervals compared to implicit dependency modeling. Importantly, the consistent gains across traffic, environment, energy, and climate datasets indicate that SGCE is not specific to a single domain: spectral decoupling generalizes to graph-structured time series with different physical mechanisms and coupling patterns. This suggests that addressing non-exchangeability in the spectral domain offers a novel perspective for reliable uncertainty quantification beyond purely time-domain adaptations.

**Ablation studies.** Table 1 also reports **SCALE w/o SGWT** and **SCALE w/o LF**. Removing LF conditioning introduces clear coverage violations, while removing SGWT weakens efficiency and stability across settings, which can be attributed to losing the spectral separation that underpins SGCE. Since our method jointly performs SGWT decomposition and low-frequency gating, it maintains reliable coverage while avoiding overly conservative intervals.

**Spectral Decoupling Analysis.** The success of SCALE can be interpreted through a spectral decoupling perspective. Following the definition of correlation intensity $c_i$ in Section 4.1, we separately visualize $c_i$ of the low- and high-frequency components. Figure 5 presents the chord diagrams, where node color indicates mean correlation intensity and edge density reflects the degree of spatial coupling. Low-frequency components exhibit consistently stronger correlations and significantly denser topological connectivity than high-frequency components. This empirical observation supports the rationale behind Definition 4.1 and provides mechanistic insight into the performance of SCALE.

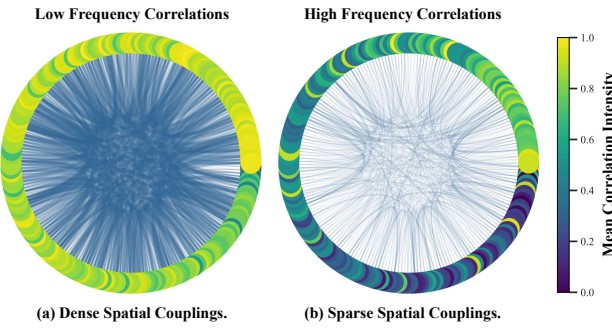

**(a) Dense Spatial Couplings.**  **(b) Sparse Spatial Couplings.**

*Figure 5.* **Spectral decoupling of spatial correlations on METR-LA.** Node color shows mean correlation while edge density reflects spatial coupling. **(a)** Low-frequency components form dense dependencies. **(b)** High-frequency components exhibit sparse connectivity, approaching weak spatial coupling for valid calibration.

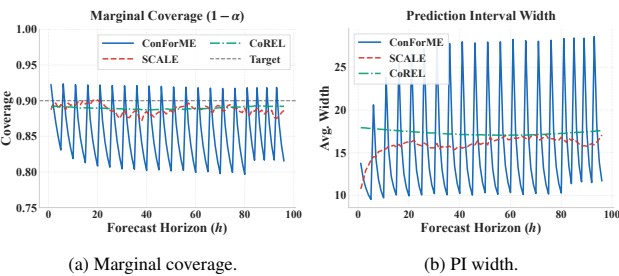

(a) Marginal coverage.  (b) PI width.

*Figure 6.* Multi-step interval diagnostics on METR-LA ($\alpha = 0.1$). We report marginal coverage and prediction interval width across horizons for SCALE, CoREL, and ConForME.

### 6.3. Performance on Multi-step Forecasting

We evaluate horizon-wise interval quality on METR-LA at $\alpha = 0.1$ using SCALE, CoREL, and ConForME in Figure 6. These methods output full-horizon quantiles in one model, enabling a clean per-horizon comparison, and single-step baselines are omitted to avoid extra design choices. Con-ForME exhibits pronounced oscillations in both coverage and width, whereas SCALE and CoREL are stable across horizons. CoREL is especially flat because it relies on a single latent state that is taken from the last encoder step and uses a shared linear readout for all horizons, which homogenizes per-step behavior. SCALE is stronger at short horizons and remains slightly better than CoREL at longer horizons with smaller widths, yielding the effectiveness of SCALE in delivering reliable long-horizon prediction intervals.

### 6.4. Parameter Sensitivity Analysis

We study the sensitivity of SCALE to the number of SGWT wavelet scales $S$, which controls the granularity of spectral decomposition. On METR-LA with $\alpha = 0.1$, we sweep $S \in \{2, 4, 6, 8, 12, 16, 24, 32, 48, 64, 96, 128, 160, 200\}$ while keeping all other settings fixed. As shown in Figure 7, SCALE maintains near-nominal coverage across a broad range of scale counts, while PI-Width and Winkler score vary only mildly. This indicates that SCALE is not

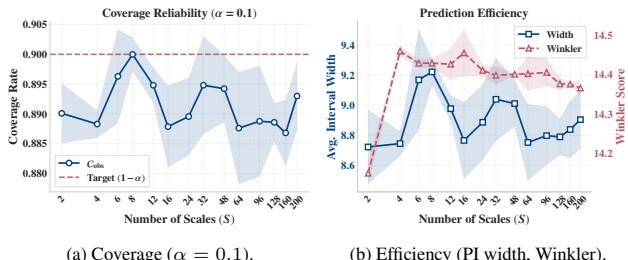

(a) Coverage ($\alpha = 0.1$).  (b) Efficiency (PI width, Winkler).

*Figure 7.* Parameter sensitivity. This figure shows the effect of the SGWT scale count $S$ on SCALE for METR-LA at $\alpha = 0.1$.

overly sensitive to the precise choice of $S$. The stable behavior suggests that the spectral conditioning effect mainly comes from low-/high-frequency separation rather than finer spectral discretization, making larger $S$ beyond a moderate range less useful. In practice, a small-to-moderate $S$ already provides a stable validity–sharpness trade-off while avoiding unnecessary computation.

## 7. Limitations

Our SCALE framework relies on spectral graph decomposition to separate strongly coupled low-frequency components from weakly coupled high-frequency components. Its reliability may therefore be reduced when this separation is weak, when residual high-frequency coupling remains strong, or when the graph structure used for decomposition is inaccurate. In such cases, the SGCE condition may only hold approximately, and the resulting coverage may deviate from the nominal level. This limitation is consistent with our theoretical analysis, where the coverage gap depends on spectral leakage and residual dependence.

## 8. Conclusion

In this paper, we introduce a novel concept termed Spectral Graph Conditional Exchangeability (SGCE) for reliable conformal prediction on graph-structured MTS. SGCE ensures exchangeability in the high-frequency components while conditioning on the low-frequency components. Building on SGCE, we propose a method named Spectral Conformal prediction via wAveLEt transform (SCALE). It leverages the spectral graph wavelet transform to derive frequency components and uses adaptive gating to condition quantile estimation on the low-frequency components. Extensive experiments across multiple domains demonstrate that SCALE achieves near-nominal coverage with competitive efficiency. Looking ahead, SCALE could be further extended to a broader range of downstream risk-sensitive decision-making problems, including intelligent transportation (Chen et al., 2024) and multi-robot planning (Wang et al., 2020). We hope our study can inspire future work on exchangeability in the spectral domain and stronger coverage guarantees for time series with complex couplings.

## Acknowledgements

This work was supported in part by the National Key Research and Development Program of China under Grant 2024YFB4708900; in part by the Natural Science Foundation of China under Grant 62225309, U24A20278, 62361166632, 62336003, 12371510, 62303307; in part by the Fundamental and Interdisciplinary Disciplines Breakthrough Plan of the Ministry of Education of China under Grant JYB2025XDXM117 and in part by the Shanghai Municipal Special Program for Basic Research on General AI Foundation Models under Grant 2025SHZDZX025G12.

## Impact Statement

This work is methodological and does not target a specific high-risk application. Nevertheless, graph-structured time series forecasting is used in decision-sensitive domains such as transportation and power systems. Since miscalibrated intervals may lead to overconfident decisions, SCALE should be deployed with spectral-decoupling diagnostics and domain-specific validation.

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

# Appendix

## A. Detailed Construction of $\mathrm{EXGAP}(\mathbf{X})$

In this appendix, we provide the detailed construction of the permutation-based exchangeability gap used in Section 4.1. Given a graph-structured multivariate time series $\mathbf{X}_{1:T} = \{\mathbf{X}_1, \ldots, \mathbf{X}_T\}$ and a block length $w > 1$, we first construct overlapping system-wide temporal blocks:

$$\mathbf{X}_j^{(w)} = \mathbf{X}_{j:j+w-1}, \qquad \mathbf{z}_j = \mathrm{vec}\left(\mathbf{X}_j^{(w)}\right), \qquad j = 1, \ldots, n, \tag{A.1}$$

where $n = T - w + 1$, $\mathbf{X}_j^{(w)} \in \mathbb{R}^{N \times w}$, and $\mathbf{z}_j \in \mathbb{R}^{Nw}$ is the vectorized representation of the $j$-th block. The collection of vectorized blocks is denoted by

$$\mathcal{B}_w(\mathbf{X}) = \{\mathbf{z}_1, \ldots, \mathbf{z}_n\}. \tag{A.2}$$

Next, let $\sigma$ be a random permutation of the temporal indices $\{1, \ldots, T\}$. The permuted sequence is written as $\mathbf{X}_\sigma = \{\mathbf{X}_{\sigma(1)}, \ldots, \mathbf{X}_{\sigma(T)}\}$. We construct the corresponding permuted temporal blocks in the same way:

$$\mathbf{X}_{j,\sigma}^{(w)} = \left[\mathbf{X}_{\sigma(j)}, \ldots, \mathbf{X}_{\sigma(j+w-1)}\right], \qquad \mathbf{z}_{j,\sigma} = \mathrm{vec}\left(\mathbf{X}_{j,\sigma}^{(w)}\right), \qquad j = 1, \ldots, n. \tag{A.3}$$

The resulting block collection is denoted by

$$\mathcal{B}_w(\mathbf{X}_\sigma) = \{\mathbf{z}_{1,\sigma}, \ldots, \mathbf{z}_{n,\sigma}\}. \tag{A.4}$$

The exchangeability gap is then defined as the expected discrepancy between the original and permuted block distributions:

$$\mathrm{EXGAP}(\mathbf{X}) = \mathbb{E}_\sigma \left[\mathrm{MMD}\left(\mathcal{B}_w(\mathbf{X}), \mathcal{B}_w(\mathbf{X}_\sigma)\right)\right], \tag{A.5}$$

where $\mathrm{MMD}(\cdot, \cdot)$ denotes the maximum mean discrepancy between two empirical distributions. In practice, the expectation over $\sigma$ is estimated by averaging over multiple random temporal permutations.

A smaller $\mathrm{EXGAP}(\mathbf{X})$ indicates that the block-level distribution is less sensitive to random temporal permutation, and is therefore empirically closer to exchangeability. Conversely, a larger value suggests stronger system-level non-exchangeability, which is typically induced by graph-based cross-node coupling and temporal dependence.

## B. Theoretical Analysis of Theorem 5.1

In this section, we provide the formal statement and proof for the validity of the SGCE framework. We proceed by first establishing the exchangeability of the non-conformity scores in the spectral domain and subsequently proving that the coverage guarantee is preserved during the reconstruction of the original graph signals.

### B.1. Formal Statement

**Theorem B.1** (Finite-Sample Coverage under SGCE). *Suppose the frequency components of the residual $\mathbf{R}_{t-W+1:t}$ satisfy SGCE (Definition 4.1). For a target miscoverage rate $\alpha \in (0,1)$, let $\widehat{\mathcal{C}}_{t+1:t+K}^{H,\alpha}$ be the prediction set constructed via split conformal prediction on the high-frequency components. Then, the marginal coverage satisfies*

$$\mathbb{P}\left(\mathbf{H}_{t+1:t+K} \in \widehat{\mathcal{C}}_{t+1:t+K}^{H,\alpha}\right) \geq 1 - \alpha. \tag{B.6}$$

*Moreover, let the final prediction set for the original signal be constructed via the deterministic mapping $\widehat{\mathcal{C}}_{t+1:t+K}^{X,\alpha} \triangleq \{\widehat{\mathbf{X}}_{t+1:t+K} + \widehat{\mathbf{L}}_{t+1:t+K} + \mathbf{h} \mid \mathbf{h} \in \widehat{\mathcal{C}}_{t+1:t+K}^{H,\alpha}\}$. Given the base prediction $\widehat{\mathbf{X}}$ and the low-frequency components $\widehat{\mathbf{L}}$ derived from the calibration procedure, this set satisfies the equivalent validity guarantee*

$$\mathbb{P}\left(\mathbf{X}_{t+1:t+K} \in \widehat{\mathcal{C}}_{t+1:t+K}^{X,\alpha}\right) \geq 1 - \alpha. \tag{B.7}$$

**B.2. Proof of Theorem 5.1**

*Proof.* The proof relies on the property that conformal validity is preserved under deterministic transformations. We structure the argument in three steps: (1) establishing exchangeability in the residual high-frequency subspace, (2) invoking the conformal lemma, and (3) demonstrating the invertibility of the spectral reconstruction.

**Step 1: Conditional Exchangeability of Scores.** Let $\mathcal{D}_{cal} = \{(\mathcal{H}_i, \mathcal{L}_i)\}_{i=1}^M$ be a calibration set of size $M$. We define a permutation-invariant non-conformity scoring function $S : \mathbb{R}^{N \times K} \to \mathbb{R}$. Let $r_i^{\mathrm{H}} = S(\mathcal{H}_i)$ denote the score computed for the $i$-th calibration unit. By SGCE (Definition 4.1), the test high-frequency $\mathcal{H}_{test}$ and the calibration residuals $\{\mathcal{H}_i\}_{i=1}^M$ are exchangeable conditional on respective low-frequency components $\mathcal{L}_i$. Consequently, the sequence of non-conformity scores $\mathcal{R} = \{r_1^{\mathrm{H}}, \ldots, r_M^{\mathrm{H}}, r_{test}^{\mathrm{H}}\}$ is also exchangeable.

**Step 2: Validity in the High-Frequency Domain.** Given the exchangeability of $\mathcal{R}$, we apply the standard lemma of split conformal prediction (Vovk et al., 2005). Let $\widehat{q}_{1-\alpha}$ be the $\lceil (M+1)(1-\alpha) \rceil / M$-th empirical quantile of the calibration scores $\{r_1^{\mathrm{H}}, \ldots, r_M^{\mathrm{H}}\}$. The probability that the test score $s_{test}$ falls within this quantile is lower-bounded by:

$$\mathbb{P}(r_{test}^{\mathrm{H}} \leq \widehat{q}_{1-\alpha}) \geq 1 - \alpha. \tag{B.8}$$

Defining the high-frequency prediction set as $\widehat{\mathcal{C}}_{t+1:t+K}^{\mathrm{H}, \alpha} = \{\mathbf{h} \mid S(\mathbf{h}) \leq \widehat{q}_{1-\alpha}\}$, this directly implies Eq. (B.6).

**Step 3: Preservation via Deterministic Reconstruction.** To recover the prediction set for the original target $\mathbf{X}_{t+1:t+K}$, we utilize the additive composition of the signal

$$\mathbf{X}_{t+1:t+K} = \widehat{\mathbf{X}}_{t+1:t+K} + \widehat{\mathbf{L}}_{t+1:t+K} + \mathbf{H}_{t+1:t+K}. \tag{B.9}$$

In the inference phase, both the base prediction $\widehat{\mathbf{X}}_{t+1:t+K}$ and the estimated low-frequency trend $\widehat{\mathbf{L}}_{t+1:t+K}$ are fixed deterministic terms (conditioned on the training/calibration data). Let $\widehat{\mathbf{\Psi}}_{t+1:t+K} = \widehat{\mathbf{X}}_{t+1:t+K} + \widehat{\mathbf{L}}_{t+1:t+K}$. The final set is constructed as the Minkowski sum $\widehat{\mathcal{C}}_{t+1:t+K}^{\mathrm{X}, \alpha} = \{\widehat{\mathbf{\Psi}}_{t+1:t+K}\} \oplus \widehat{\mathcal{C}}_{t+1:t+K}^{\mathrm{H}, \alpha}$. Crucially, the event of the true signal falling into the prediction set is logically equivalent to the high-frequency residual falling into its corresponding set:

$$\mathbf{X}_{t+1:t+K} \in \widehat{\mathcal{C}}_{t+1:t+K}^{\mathrm{X}, \alpha} \iff \widehat{\mathbf{\Psi}}_{t+1:t+K} + \mathbf{H}_{t+1:t+K} \in \{\widehat{\mathbf{\Psi}}_{t+1:t+K} + \mathbf{h} \mid \mathbf{h} \in \widehat{\mathcal{C}}_{t+1:t+K}^{\mathrm{H}, \alpha}\}$$
$$\iff \mathbf{H}_{t+1:t+K} \in \widehat{\mathcal{C}}_{t+1:t+K}^{\mathrm{H}, \alpha}. \tag{B.10}$$

Since the events are identical, their probabilities are equal. Substituting the bound from Step 2 yields the final guarantee:

$$\mathbb{P}\left(\mathbf{X}_{t+1:t+K} \in \widehat{\mathcal{C}}_{t+1:t+K}^{\mathrm{X}, \alpha}\right) = \mathbb{P}\left(\mathbf{H}_{t+1:t+K} \in \widehat{\mathcal{C}}_{t+1:t+K}^{\mathrm{H}, \alpha}\right) \geq 1 - \alpha. \tag{B.11}$$

This concludes the proof. $\square$

# C. Theoretical Analysis of Theorem 5.2

In this section, we provide the formal statement and detailed derivation of the robust coverage guarantee presented in Theorem 5.2. We explicitly quantify the finite-sample validity deviations arising from the spectral graph wavelet transform, characterizing them in terms of wavelet approximation errors and residual spatial dependencies.

**C.1. Formal Statement**

We first provide the rigorous version of Theorem 5.2, which explicitly characterizes the coverage gap $\delta$.

**Theorem C.1** (Robust Finite-Sample Coverage). *Consider the setup of Theorem 5.2. Let $\widehat{\mathcal{L}}_t$ be the estimated low-frequency sequences and $\mathcal{L}_t$ be the ground truth. Assume the non-conformity scoring function $\tilde{S}(\cdot, \cdot)$ is $C_s$-Lipschitz continuous with respect to both arguments. The prediction set $\widehat{\mathcal{C}}_{t+1:t+K}^{sgwt, \alpha}$ satisfies the following robust marginal coverage bound*

$$\mathbb{P}(\mathbf{X}_{t+1:t+K} \in \widehat{\mathcal{C}}_{t+1:t+K}^{sgwt, \alpha}) \geq 1 - \alpha - \left(C_e C_s \cdot \mathbb{E}[\|\mathcal{L}_t - \widehat{\mathcal{L}}_t\|] + \delta_{dep}\right), \tag{C.12}$$

*where:*

- $\mathbb{E}[\|\mathcal{L}_t - \widehat{\mathcal{L}}_t\|]$ *represents the expected spectral approximation error, e.g., due to polynomial wavelet truncation.*

- $C_e$ denotes the Lipschitz constant of the cumulative distribution function (CDF) on the score distribution.

- $\delta_{dep}$ quantifies the deviation from exchangeability caused by residual spatial coupling, measured by the Total Variation distance between the true residual process and an idealized exchangeable process.

## C.2. Proof of Theorem C.1

*Proof.* Let $\mathcal{D}_{cal} = \{(\mathcal{H}_i, \mathcal{L}_i)\}_{i=1}^M$ be a calibration set of size $M$. Let $\Pi_P(r)$ denote the distribution of the observed scores (computed using estimated components) under the true data distribution $P$. Let $\Pi_Q(\tilde{r})$ denote the distribution of the ideal scores (computed using true components) under the idealized reference distribution $Q$ where residuals are exchangeable. We aim to bound the coverage gap by analyzing the Total Variation (TV) distance between these two laws:

$$\delta \triangleq d_{\text{TV}}(\Pi_P(r), \Pi_Q(\tilde{r})). \tag{C.13}$$

By the triangle inequality for the TV metric, we can decompose this distance into a stability term and a dependence term:

$$d_{\text{TV}}(\Pi_P(r), \Pi_Q(\tilde{r})) \leq \underbrace{d_{\text{TV}}(\Pi_P(r), \Pi_P(\tilde{r}))}_{\text{Spectral Leakage Gap } (\delta_{\text{leak}})} + \underbrace{d_{\text{TV}}(\Pi_P(\tilde{r}), \Pi_Q(\tilde{r}))}_{\text{Dependence Gap } (\delta_{\text{dep}})}. \tag{C.14}$$

**Step 1: The Ideal Reference Case.** Consider the ideal non-conformity scores derived from the true spectral components

$$\tilde{r}_i = \tilde{S}(\mathcal{H}_i, \mathcal{L}_i), \quad \forall i \in \mathcal{D}_{cal} \cup \{test\}. \tag{C.15}$$

Under the ideal distribution $Q$, Theorem 5.1 ensures that the sequence of scores is exchangeable. Consequently, standard conformal prediction guarantees hold

$$\mathbb{Q}(\tilde{r}_{\text{test}} \leq \widehat{q}_{1-\alpha}) \geq 1 - \alpha, \tag{C.16}$$

where $\widehat{q}_{1-\alpha}$ is the empirical quantile of the calibration scores.

**Step 2: Stability under Spectral Approximation.** In practice, we compute scores using the estimated high-frequency components $\widehat{\mathbf{H}}_t = \mathbf{X}_t - \widehat{\mathbf{L}}_t = \mathbf{H}_t + \mathbf{L}_t - \widehat{\mathbf{L}}_t$ and low-frequency components $\widehat{\mathbf{L}}_t$. The observed score is

$$r_i = \tilde{S}(\widehat{\mathcal{H}}_i, \widehat{\mathcal{L}}_i) = \tilde{S}(\mathcal{H}_i + (\mathcal{L}_i - \widehat{\mathcal{L}}_i), \widehat{\mathcal{L}}_i). \tag{C.17}$$

Given the $C_s$-Lipschitz continuity of $\tilde{S}(\cdot, \cdot)$, the pointwise error is bounded by

$$|r_i - \tilde{r}_i| \leq C_s \left( \|\widehat{\mathcal{H}}_i - \mathcal{H}_i\| + \|\widehat{\mathcal{L}}_i - \mathcal{L}_i\| \right) \leq 2C_s \|\mathcal{L}_i - \widehat{\mathcal{L}}_i\|. \tag{C.18}$$

*(Note: We absorb the factor of 2 into $C_s$ for notational simplicity in the final bound).*

To map this value perturbation to a probability shift, we assume the CDF of the scores is $C_e$-Lipschitz. Denoting the laws of $r, \tilde{r}$ as $\Pi_P(r), \Pi_P(\tilde{r})$, the Wasserstein distance between these laws is bounded by the Total Variation (TV) distance

$$\delta_{\text{leak}} \triangleq d_{\text{TV}}(\Pi_P(r), \Pi_P(\tilde{r})) \leq C_e C_s \cdot \mathbb{E}[\|\mathcal{L}_t - \widehat{\mathcal{L}}_t\|]. \tag{C.19}$$

**Step 3: Bounding Spatial Dependence.** Even with perfect filters, the true high-frequency residuals may exhibit weak residual dependencies. We define $\delta_{\text{dep}}$ as the TV distance between the joint distribution of the actual residuals $\Pi_P(\tilde{r})$ and the closest exchangeable product measure $\Pi_Q(\tilde{r})$, namely

$$\delta_{\text{dep}} \triangleq d_{\text{TV}}(\Pi_P(\tilde{r}), \Pi_Q(\tilde{r})). \tag{C.20}$$

**Step 4: Synthesis via Triangle Inequality.** We seek to lower bound the coverage under the true distribution $P$. Using the triangle inequality on the TV distances derived in (C.14). Since $\mathbb{Q}(\tilde{r}_{\text{test}} \leq \widehat{q}_{1-\alpha}) \geq 1 - \alpha$ in (C.16), we have:

$$\begin{aligned} \mathbb{P}(r_{\text{test}} \leq \widehat{q}_{1-\alpha}) &\geq \mathbb{Q}(\tilde{r}_{\text{test}} \leq \widehat{q}_{1-\alpha}) - d_{\text{TV}}(\Pi_P(r), \Pi_Q(\tilde{r})) \\ &\geq (1 - \alpha) - (\delta_{\text{leak}} + \delta_{\text{dep}}) \\ &\geq 1 - \alpha - \left( C_e C_s \mathbb{E}[\|\mathcal{L}_t - \widehat{\mathcal{L}}_t\|] + \delta_{\text{dep}} \right). \end{aligned} \tag{C.21}$$

This completes the proof. $\qquad\square$

# D. SGWT Cutoff Auto-Selection

---

**Algorithm 1** SGWT cutoff auto-selection

---

**Require:** adjacency matrix $\mathbf{A}$, signal samples $\mathbf{X}$, max scales $S$, kernel type, max samples $T_{\max}$, optional correlation threshold $\tau$

**Ensure:** suggested high-frequency scale count $k$

1: Flatten $\mathbf{X}$ to $(T, N)$ and subsample to at most $T_{\max}$ rows.
2: Initialize SGWT with $\mathbf{A}$; obtain kernels $g(\cdot)$, scale list $\{s_i\}_{i=0}^{S-1}$, and eigenvectors $\mathbf{U}$.
3: Transform to the spectral domain: $\widehat{\mathbf{X}} = \mathbf{X}\mathbf{U}$.
4: **for** each scale $s_i$ **do**
5: $\quad\widehat{\mathbf{W}}_i = \widehat{\mathbf{X}} \odot g(s_i, \boldsymbol{\lambda})$.
6: $\quad\mathbf{W}_i = \widehat{\mathbf{W}}_i \mathbf{U}^\top$.
7: $\quad$Record (i) spatial correlation (mean absolute off-diagonal), (ii) distribution discrepancy (mean KS over random node pairs), (iii) energy (mean square).
8: **end for**
9: Smooth the correlation curve with a short kernel.
10: **if** $\tau$ is given **then**
11: $\quad$Set $k$ to the first index with smoothed correlation $> \tau$.
12: **else**
13: $\quad$Compute candidate cut points: first index where cumulative energy $\geq 0.9$, steepest correlation increase, and maximal negative curvature.
14: $\quad$Set $k$ to the median of candidates.
15: **end if**
16: Clamp $k$ to $[1, S]$ and return.

---

# E. Experimental Setup and Hyperparameters

This section reports the experimental setup in an appendix style consistent with related forecasting and conformal works.

## E.1. Hardware and Software

All experiments are implemented in Python (Python Software Foundation, 2026) using standard scientific and deep-learning libraries, including NumPy (Harris et al., 2020), PyTorch (Paszke et al., 2019), PyTorch Lightning (Falcon & The PyTorch Lightning Team, 2025), and PyTorch Geometric (Fey & Lenssen, 2019). We use Torch Spatio-temporal for spatio-temporal data handling (Cini & Marisca, 2022). Baseline reference implementations are publicly available, including HopCPT[1], CoREL[2], ConForME[3], and EnbPI[4].

## E.2. Datasets and Splits

We evaluate on graph-structured multivariate time series benchmarks spanning multiple domains: METR-LA (traffic), AirQuality (environment), Electricity (energy), and USHCN-West (climate) form the cross-domain main evaluation reported in the main text. PEMS04, PEMS07, and PEMS08 are additionally used for traffic-only benchmarking and ablations in Appendix F.1 and Appendix F.2. We adopt a temporal split with train/calibration/test ratios of 0.4/0.4/0.2. All interval methods operate on the same residual tensors produced by the first stage backbone and share the same calibration/test split. For trainable interval predictors, we further reserve a small validation subset (10% of the calibration portion) for early stopping.

---

[1] https://github.com/ml-jku/HopCPT
[2] https://github.com/andreacini/corel
[3] https://github.com/aloysiogl/conforme
[4] https://github.com/hamrel-cxu/EnbPI

| Dataset | Region/Source | Time span | Interval | Nodes | Edges | Graph construction |
|---------|--------------|-----------|----------|-------|-------|--------------------|
| METR-LA | Los Angeles County | 2012-03–2012-06 | 5-min | 207 | 674 | Distance-based adjacency (Gaussian kernel) |
| AirQuality | 437 monitoring stations | 2014-05–2015-04 | 1-hour | 437 | 2,699 | Distance-based adjacency (Gaussian kernel) |
| Electricity | 370 customers, Portugal | 2011-01–2014-12 | 1-hour | 370 | – | Correlation-based adjacency |
| USHCN-West | USHCN western US | 1888–2014 | daily | 218 | – | Distance-based adjacency (Gaussian kernel) |
| PEMS04 | District 4, CA | 2018-01–2018-02 | 5-min | 307 | 104 | Distance-based adjacency (Gaussian kernel) |
| PEMS07 | District 7, CA | 2017-05–2017-08 | 5-min | 883 | 395 | Distance-based adjacency (Gaussian kernel) |
| PEMS08 | District 8, CA | 2016-07–2016-08 | 5-min | 170 | 68 | Distance-based adjacency (Gaussian kernel) |

*Table 2.* Datasets used in this work. The top block contains the cross-domain datasets (traffic, environment, energy, and climate) used for the main results in the main text. The bottom block contains additional traffic datasets used for the appendix studies and the multi-step / parameter sensitivity analyses.

| Item | Setting |
|------|---------|
| Split | temporal 0.4/0.4/0.2 (train/calibration/test) |
| Window / horizon | $W = 12$, $K = 1$ (main); $K = 96$ (multi-step on METR-LA) |
| Backbone | GRU + DiffConv template (2 message-passing layers) |
| Hidden / embedding | 32 / 16 |
| Optimizer | Adam ($3 \times 10^{-3}$, weight decay 0) |
| Training | 200 epochs, batch size 32, 100 train batches per epoch |
| Early stopping | patience 50 on the validation portion of the temporal split |
| Scaling | standard scaling across nodes (graph axis) |

*Table 3.* The first stage backbone configuration used to generate residuals.

### E.3. Dataset Details

We summarize dataset statistics following the Torch Spatiotemporal dataset documentation[5]. Graph edges are computed from the provided distance matrices using a thresholded Gaussian kernel with $\theta = 0.1$ and no self-loops (matching the distance-based connectivity used in our configs); edges are reported as undirected nonzero weights.

### E.4. The first stage Backbone and Residual Generation

**Backbone architecture.** The point forecaster uses a GRU + DiffConv template with one GRU-based temporal layer and two graph message-passing layers, using hidden size 32 and node embedding size 16 with elu activation. This backbone is directly adopted from CoREL for aligned comparison, and Appendix F.2 reports results with alternative backbones (Transformer, GWNet, DCRNN). The architecture consists of a linear input encoder, a GRU temporal encoder, stacked DiffConv layers, and a linear readout. The adjacency is constructed from distance-based connectivity with threshold 0.1 and without self-loops.

**Input/output protocol.** Unless otherwise stated, we use an input window of $W = 12$ and single-step forecasting horizon $H = 1$ with stride 1. For the multi-step analysis, we set $H = 96$ on METR-LA.

**Optimization.** The backbone is trained with Adam (learning rate $3 \times 10^{-3}$, weight decay 0) for up to 200 epochs, batch size 32, and gradient clipping 5. Inputs are standardized (graph-level standard scaling) and exogenous time features (time-of-day and day-of-week) are included when available.

### E.5. Evaluation Protocol

We evaluate coverage and efficiency at $\alpha \in \{0.05, 0.1, 0.2\}$. We report empirical **Coverage**, **PI-Width** and **Winkler** (lower is better for efficiency metrics). In the main table, a coverage checkmark indicates an absolute coverage error no larger than 0.02.

Trainable interval predictors are run with five random seeds (0–4) and we report mean±std. Non-trainable baselines are deterministic given residuals and are run once.

---

[5] https://torch-spatiotemporal.readthedocs.io/en/latest/modules/datasets.html

## E.6. Hyperparameters and Experimental Setup

Hyperparameters are either taken from the corresponding reference implementations or tuned on a validation subset of the calibration data, following common appendix styles in conformal time-series works. All interval baselines are applied to the same residual tensors produced by the first stage point forecaster and share the same temporal calibration/test split. For trainable interval predictors, we further reserve a validation subset from the calibration portion for early stopping (method-specific choices are stated below).

**SCP.** SCP (Vovk et al., 2005) is the standard split-conformal baseline: it calibrates empirical quantiles of residual-based nonconformity scores on the calibration split and forms two-sided prediction intervals by adding the corresponding lower/upper residual quantiles to the point forecasting, for each $\alpha \in \{0.05, 0.1, 0.2\}$.

**SeqCP.** SeqCP (Xu & Xie, 2023b) is a sequential conformal variant that adapts to temporal dependence and drift by recalibrating on a recent window of residuals. We use a sliding-window procedure with window length $M_{\text{seq}} = 100$, i.e., the interval quantiles are computed from the most recent $M_{\text{seq}}$ residuals in the calibration stream.

**NexCP.** NexCP (Barber et al., 2023) performs conformal calibration using exponentially decaying weights over past residuals, which emphasizes recent observations while still leveraging a longer history. We use decay parameter $\rho = 0.99$ when computing the (weighted) residual quantiles.

**EnbPI.** EnbPI (Xu & Xie, 2023a) constructs prediction intervals from an ensemble of bootstrap predictors and an online residual buffer that is updated over time; at each step, intervals are formed by combining the ensemble prediction with empirical quantiles of the residual buffer. We follow the official implementation and use the default protocol throughout all datasets: $B = 30$ bootstrap models, stride 1, lag 20, and RidgeCV as the base regressor. We evaluate EnbPI in the oracle online-update setting (i.e., using true residuals for online updates), and do not introduce any dataset-specific changes beyond the original implementation.

**HopCPT.** HopCPT (Auer et al., 2023) employs a Hopfield-style associative memory to calibrate node-wise quantiles over time. We follow the reference configuration in the original implementation: original mode with batch size 4 time series, trained for $n_{\text{epochs}} = 3000$ using AdamW with learning rate 0.01, weight decay 0.001, $(\beta_1, \beta_2) = (0.9, 0.999)$, and $\epsilon = 0.01$. We validate every 5 epochs and use early-stopping patience 10000 (effectively disabling early stopping) to match the reference training protocol.

**ConForME.** ConForME (Galvão Lopes et al., 2024) constructs multi-horizon prediction intervals by conformalizing a learned interval predictor via conditional calibration. We use the conforme variant with approximate partition size 5 and train for 100 epochs with learning rate $10^{-6}$. We enable the quantile-offset optimization with binary-search tolerance 0.01 (as in the reference implementation) to reduce interval width while targeting the same nominal coverage.

**CoREL.** CoREL (Cini et al., 2025) fits a *relational quantile predictor* that models cross-series dependence via a latent graph and estimates a discrete grid of conditional residual quantiles; two-sided prediction intervals are obtained by selecting the appropriate lower/upper quantiles from this grid. We follow the original paper *exactly* and use the same training and model-selection procedure (including tuning the GRU + DiffConv template width on 10% of the calibration data). For real-world datasets, the model is trained on the calibration set for a maximum of 100 epochs, where each epoch consists of at most 50 mini-batches of size 64. We use Adam with initial learning rate 0.003 and a stepwise decay that reduces the learning rate by 75% every 20 epochs, and we set the graph neighborhood size to $K = 20$. Following the reference implementation, the readout predicts 39 equally spaced quantiles and we use the first 25% of the calibration split as a validation set for early stopping.

**SCALE.** We use SGWT with $S = 4$ wavelet scales and mexican-hat kernels, together with the adaptive gated conditioning mechanism. Training uses learning rate $8 \times 10^{-4}$, weight decay $10^{-4}$, and MultiStepLR with gamma 0.5 and milestones at epochs 5 and 8. We train for 30 epochs on datasets with batch size 128. The crossing penalty is set to 15 on all datasets.

### E.7. Computational Cost

We further report the computational cost of SCALE to clarify its practical overhead. SCALE introduces spectral operations through SGWT and a data-driven cutoff selection procedure, but these operations incur only a one-time preprocessing cost. Specifically, the spectral cutoff is selected once for each dataset and then reused for all subsequent training and evaluation runs. After this preprocessing step, SCALE operates with fixed spectral filters and only adds a lightweight low-frequency encoder, a parameter-free high-frequency statistics extractor, and an adaptive gating module.

Table 4 reports the wall-clock runtime of different interval construction methods under the same residual-interval evaluation protocol. The results show that SCALE remains computationally efficient compared with trainable or online adaptive conformal baselines. Although SCALE is more expensive than simple empirical-quantile baselines such as SCP and SeqCP, it is substantially faster than CoREL, NexCP, EnbPI, HopCPT, and ConForME on most datasets, while achieving a better coverage–efficiency trade-off.

*Table 4.* Runtime comparison of different conformal prediction methods. Runtime is reported in seconds under the same residual-interval evaluation protocol.

| Method | METR-LA | PEMS07 | PEMS08 | PEMS04 |
|---|---|---|---|---|
| SCP | 10.0 | 2.4 | 1.2 | 1.2 |
| SeqCP | 9.6 | 25.2 | 4.4 | 6.6 |
| CoREL | 843.9 | 183.5 | 46.4 | 48.2 |
| NexCP | 2217.2 | 10048.5 | 454.7 | 775.0 |
| EnbPI | 17046.0 | 6816.2 | 683.0 | 1009.3 |
| HopCPT | 186996.3 | 417650.7 | 272850.6 | 254422.4 |
| ConForME | 632.0 | 2333.0 | 338.0 | 120.8 |
| SCALE | 238.4 | 292.0 | 40.2 | 99.3 |

## F. Additional Results

This section provides complementary results supporting the main claims.

### F.1. Results on Traffic Benchmarks

For completeness and to align with prior benchmarks in spatio-temporal graph forecasting, we also report the residual-interval results on the four widely used traffic datasets METR-LA, PEMS07, PEMS08, and PEMS04. The setup follows Section F.2, with a GRU + DiffConv template backbone, $W = 12$, $K = 1$, the same temporal split, and the same set of baselines. Table 5 demonstrates that SCALE consistently maintains near-nominal coverage and produces sharper intervals across miscoverage levels $\alpha \in \{0.05, 0.1, 0.2\}$ on these traffic benchmarks, complementing the cross-domain results reported in the main text.

### F.2. Robustness to the first stage Backbone

To highlight the model-free nature of our conformal layer, we additionally repeat the main interval evaluation by replacing the first stage point forecaster with standard spatio-temporal forecasting backbones implemented in the Torch Spatiotemporal library (tsl). We include Transformer, GWNet, and DCRNN in this robustness study. All conformal methods operate on the resulting residual tensors and use the same calibration/test split and evaluation protocol. Tables 7, 8, and 9 summarize the results for different backbones.

**Backbone architecture.** We summarize the first stage backbones used to generate residuals in Table 6. For non-template backbones (e.g., Transformer/GWNet/DCRNN), we use the default tsl implementations and their standard hyperparameters.

### F.3. Compatibility with Stronger Forecasting Backbones

To further examine the compatibility of SCALE with stronger modern forecasting backbones, we replace the default forecaster with PatchSTG (Fang et al., 2024), MAGE (Ma et al., 2025), and D2STGNN (Shao et al., 2022b). SCALE is a post-hoc conformal calibration framework: its second stage operates on the residual representation produced by the first-stage forecaster, rather than depending on the internal architecture of the forecasting model. Table 10 reports the results on both traffic and non-traffic datasets at $\alpha = 0.1$. The results show that SCALE remains broadly compatible with stronger

*Table 5.* Residual-interval results using a GRU + DiffConv template backbone on the four traffic datasets METR-LA, PEMS07, PEMS08, and PEMS04 at $\alpha \in \{0.05, 0.1, 0.2\}$. Coverage, PI-Width, and Winkler are reported as mean±std across seeds for trainable methods.

| Method | METR-LA | | | PEMS07 | | | PEMS08 | | | PEMS04 | | |
|---|---|---|---|---|---|---|---|---|---|---|---|---|
| | Coverage ↑ | PI-Width ↓ | Winkler ↓ | Coverage ↑ | PI-Width ↓ | Winkler ↓ | Coverage ↑ | PI-Width ↓ | Winkler ↓ | Coverage ↑ | PI-Width ↓ | Winkler ↓ |
| **$\alpha = 0.05$** | | | | | | | | | | | | |
| SCP (Vovk et al., 2005) | 0.9470 ± 0.0000 ✓ | 13.62 ± 0.00 | 21.98 ± 0.00 | 0.9459 ± 0.0000 ✓ | 95.52 ± 0.00 | 139.31 ± 0.00 | 0.9493 ± 0.0000 ✓ | 72.59 ± 0.00 | 102.42 ± 0.00 | 0.9533 ± 0.0000 ✓ | 101.38 ± 0.00 | 135.44 ± 0.00 |
| SeqCP (Xu & Xie, 2023b) | 0.9097 ± 0.0000 ✗ | 12.02 ± 0.00 | 22.24 ± 0.00 | 0.9195 ± 0.0000 ✗ | 88.21 ± 0.00 | 139.82 ± 0.00 | 0.9202 ± 0.0000 ✗ | 66.15 ± 0.00 | 104.98 ± 0.00 | 0.9067 ± 0.0000 ✗ | 86.36 ± 0.00 | 141.68 ± 0.00 |
| CoREL (Cini et al., 2025) | 0.9509 ± 0.0041 ✓ | 12.19 ± 0.44 | 18.10 ± 0.24 | 0.9497 ± 0.0009 ✓ | 86.08 ± 0.91 | 115.58 ± 0.93 | 0.9504 ± 0.0012 ✓ | 64.55 ± 0.73 | 84.58 ± 0.48 | 0.9538 ± 0.0028 ✓ | 84.22 ± 1.12 | **108.33 ± 0.23** |
| NexCP (Barber et al., 2023) | 0.9433 ± 0.0000 ✓ | 13.33 ± 0.00 | 21.20 ± 0.00 | 0.9452 ± 0.0000 ✓ | 94.63 ± 0.00 | 132.92 ± 0.00 | 0.9458 ± 0.0000 ✓ | 70.98 ± 0.00 | 99.02 ± 0.00 | 0.9438 ± 0.0000 ✓ | 95.35 ± 0.00 | 132.17 ± 0.00 |
| EnbPI (Xu & Xie, 2023a) | 0.9488 ± 0.0000 ✓ | 13.65 ± 0.00 | 21.90 ± 0.00 | 0.9485 ± 0.0000 ✓ | 95.19 ± 0.00 | 136.59 ± 0.00 | 0.9501 ± 0.0000 ✓ | 72.02 ± 0.02 | 101.08 ± 0.01 | 0.9509 ± 0.0000 ✓ | 98.98 ± 0.01 | 134.52 ± 0.01 |
| HopCPT (Auer et al., 2023) | 0.9349 ± 0.0025 ✓ | 13.16 ± 0.35 | 21.32 ± 0.60 | 0.9301 ± 0.0254 ✓ | 97.46 ± 3.57 | 144.38 ± 19.38 | 0.9432 ± 0.0012 ✓ | 70.76 ± 2.24 | 99.95 ± 1.49 | 0.9425 ± 0.0012 ✓ | 94.18 ± 1.20 | 131.32 ± 2.45 |
| ConForME (Galvão Lopes et al., 2024) | 0.9471 ± 0.0000 ✓ | 13.84 ± 0.00 | 22.61 ± 0.00 | 0.9463 ± 0.0000 ✓ | 95.89 ± 0.00 | 139.56 ± 0.00 | 0.9496 ± 0.0000 ✓ | 72.89 ± 0.00 | 102.92 ± 0.00 | 0.9529 ± 0.0000 ✓ | 101.48 ± 0.00 | 136.33 ± 0.00 |
| SCALE w/o LF | 0.9473 ± 0.0010 ✓ | 13.22 ± 0.07 | 20.14 ± 0.01 | 0.9370 ± 0.0010 ✓ | 96.15 ± 0.55 | 151.31 ± 0.19 | 0.9207 ± 0.0027 ✗ | 60.15 ± 0.75 | 105.46 ± 0.56 | 0.8995 ± 0.0049 ✗ | 70.15 ± 1.44 | 152.71 ± 2.04 |
| SCALE w/o SGWT | 0.9448 ± 0.0061 ✓ | 11.99 ± 0.43 | 18.76 ± 0.06 | 0.9514 ± 0.0063 ✓ | 88.63 ± 3.21 | 118.84 ± 0.36 | 0.9497 ± 0.0087 ✓ | 64.84 ± 2.85 | 86.20 ± 0.20 | 0.9579 ± 0.0041 ✓ | 89.73 ± 2.25 | 113.23 ± 0.47 |
| SCALE | 0.9436 ± 0.0030 ✓ | **11.69 ± 0.23** | **17.92 ± 0.07** | 0.9481 ± 0.0023 ✓ | **81.64 ± 0.92** | **115.09 ± 0.11** | 0.9551 ± 0.0023 ✓ | **61.16 ± 0.90** | **81.63 ± 0.14** | 0.9486 ± 0.0023 ✓ | **82.80 ± 0.92** | 109.66 ± 0.10 |
| **$\alpha = 0.10$** | | | | | | | | | | | | |
| SCP (Vovk et al., 2005) | 0.8950 ± 0.0000 ✓ | 9.46 ± 0.00 | 16.79 ± 0.00 | 0.8947 ± 0.0000 ✓ | 73.88 ± 0.00 | 111.84 ± 0.00 | 0.9003 ± 0.0000 ✓ | 56.27 ± 0.00 | 82.50 ± 0.00 | 0.9055 ± 0.0000 ✓ | 78.82 ± 0.00 | 110.69 ± 0.00 |
| SeqCP (Xu & Xie, 2023b) | 0.8554 ± 0.0000 ✗ | 8.79 ± 0.00 | 16.65 ± 0.00 | 0.8677 ± 0.0000 ✗ | 70.16 ± 0.00 | 111.52 ± 0.00 | 0.8682 ± 0.0000 ✗ | 52.77 ± 0.00 | 83.68 ± 0.00 | 0.8524 ± 0.0000 ✗ | 69.23 ± 0.00 | 113.52 ± 0.00 |
| CoREL (Cini et al., 2025) | 0.9031 ± 0.0084 ✓ | 9.15 ± 0.34 | 14.30 ± 0.14 | 0.8990 ± 0.0034 ✓ | 69.64 ± 0.78 | 96.29 ± 0.67 | 0.8987 ± 0.0038 ✓ | 52.25 ± 0.61 | 71.02 ± 0.37 | 0.9056 ± 0.0039 ✓ | 68.91 ± 0.70 | **91.50 ± 0.10** |
| NexCP (Barber et al., 2023) | 0.8934 ± 0.0000 ✓ | 9.33 ± 0.00 | 16.07 ± 0.00 | 0.8953 ± 0.0000 ✓ | 73.49 ± 0.00 | 107.59 ± 0.00 | 0.8966 ± 0.0000 ✓ | 55.35 ± 0.00 | 80.32 ± 0.00 | 0.8935 ± 0.0000 ✓ | 74.37 ± 0.00 | 107.96 ± 0.00 |
| EnbPI (Xu & Xie, 2023a) | 0.8986 ± 0.0000 ✓ | 9.45 ± 0.00 | 16.67 ± 0.00 | 0.8978 ± 0.0000 ✓ | 73.75 ± 0.00 | 110.25 ± 0.00 | 0.9004 ± 0.0000 ✓ | 55.90 ± 0.01 | 81.77 ± 0.01 | 0.9021 ± 0.0000 ✓ | 77.08 ± 0.01 | 110.01 ± 0.00 |
| HopCPT (Auer et al., 2023) | 0.8832 ± 0.0040 ✓ | 9.84 ± 0.13 | 16.52 ± 0.40 | 0.8822 ± 0.0245 ✓ | 78.22 ± 5.18 | 115.98 ± 13.61 | 0.8937 ± 0.0010 ✓ | 56.04 ± 2.22 | 81.27 ± 1.72 | 0.8925 ± 0.0015 ✓ | 74.48 ± 1.31 | 107.51 ± 0.98 |
| ConForME (Galvão Lopes et al., 2024) | 0.8955 ± 0.0000 ✓ | 9.34 ± 0.00 | 17.09 ± 0.00 | 0.8951 ± 0.0000 ✓ | 74.30 ± 0.00 | 112.08 ± 0.00 | 0.9002 ± 0.0000 ✓ | 56.50 ± 0.00 | 82.85 ± 0.00 | 0.9049 ± 0.0000 ✓ | 78.83 ± 0.00 | 111.31 ± 0.00 |
| SCALE w/o LF | 0.8968 ± 0.0019 ✓ | 9.56 ± 0.07 | 15.72 ± 0.00 | 0.8988 ± 0.0003 ✓ | 78.38 ± 0.11 | 119.84 ± 0.06 | 0.8968 ± 0.0013 ✓ | 53.95 ± 0.27 | 82.56 ± 0.07 | 0.8829 ± 0.0033 ✓ | **65.32 ± 0.83** | 112.39 ± 0.40 |
| SCALE w/o SGWT | 0.8939 ± 0.0106 ✓ | **8.88 ± 0.37** | 14.66 ± 0.02 | 0.9029 ± 0.0105 ✓ | 71.21 ± 2.51 | 98.34 ± 0.25 | 0.8990 ± 0.0137 ✓ | 52.27 ± 2.34 | 72.11 ± 0.12 | 0.9144 ± 0.0074 ✓ | 72.71 ± 1.80 | 94.88 ± 0.35 |
| SCALE | 0.8913 ± 0.0039 ✓ | 8.89 ± 0.16 | **14.27 ± 0.03** | 0.9015 ± 0.0038 ✓ | **66.25 ± 0.76** | **94.47 ± 0.08** | 0.9098 ± 0.0048 ✓ | **49.41 ± 0.77** | **67.92 ± 0.11** | 0.9005 ± 0.0036 ✓ | 67.63 ± 0.76 | 91.97 ± 0.07 |
| **$\alpha = 0.20$** | | | | | | | | | | | | |
| SCP (Vovk et al., 2005) | 0.7933 ± 0.0000 ✓ | 5.86 ± 0.00 | 12.18 ± 0.00 | 0.7941 ± 0.0000 ✓ | 53.13 ± 0.00 | 87.30 ± 0.00 | 0.8017 ± 0.0000 ✓ | 40.38 ± 0.00 | 64.65 ± 0.00 | 0.8076 ± 0.0000 ✓ | 55.74 ± 0.00 | 87.27 ± 0.00 |
| SeqCP (Xu & Xie, 2023b) | 0.7573 ± 0.0000 ✗ | 5.87 ± 0.00 | 11.99 ± 0.00 | 0.7726 ± 0.0000 ✗ | 51.84 ± 0.00 | 86.78 ± 0.00 | 0.7741 ± 0.0000 ✗ | 39.11 ± 0.00 | 65.13 ± 0.00 | 0.7572 ± 0.0000 ✗ | 51.08 ± 0.00 | 88.27 ± 0.00 |
| CoREL (Cini et al., 2025) | 0.8061 ± 0.0054 ✓ | 6.57 ± 0.11 | **10.96 ± 0.06** | 0.7982 ± 0.0064 ✓ | 52.63 ± 0.68 | 78.26 ± 0.49 | 0.7978 ± 0.0041 ✓ | 39.71 ± 0.41 | 58.20 ± 0.30 | 0.8080 ± 0.0053 ✓ | 52.66 ± 0.56 | 75.37 ± 0.06 |
| NexCP (Barber et al., 2023) | 0.7930 ± 0.0000 ✓ | 5.95 ± 0.00 | 11.67 ± 0.00 | 0.7961 ± 0.0000 ✓ | 53.18 ± 0.00 | 84.81 ± 0.00 | 0.7974 ± 0.0000 ✓ | 40.07 ± 0.00 | 63.50 ± 0.00 | 0.7935 ± 0.0000 ✓ | 53.10 ± 0.00 | 85.40 ± 0.00 |
| EnbPI (Xu & Xie, 2023a) | 0.7986 ± 0.0000 ✓ | 5.87 ± 0.00 | 12.05 ± 0.00 | 0.7973 ± 0.0000 ✓ | 52.94 ± 0.00 | 86.38 ± 0.00 | 0.8008 ± 0.0001 ✓ | 40.02 ± 0.00 | 64.25 ± 0.00 | 0.8029 ± 0.0000 ✓ | 54.62 ± 0.00 | 86.83 ± 0.00 |
| HopCPT (Auer et al., 2023) | 0.7796 ± 0.0067 ✗ | 6.88 ± 0.15 | 12.46 ± 0.25 | 0.7857 ± 0.0219 ✓ | 58.16 ± 5.11 | 91.46 ± 9.77 | 0.7937 ± 0.0010 ✓ | 40.98 ± 2.37 | 64.54 ± 1.99 | 0.7916 ± 0.0021 ✓ | 54.02 ± 2.24 | 85.60 ± 0.75 |
| ConForME (Galvão Lopes et al., 2024) | 0.7939 ± 0.0000 ✓ | **5.80 ± 0.00** | 12.28 ± 0.00 | 0.7947 ± 0.0000 ✓ | 53.53 ± 0.00 | 87.55 ± 0.00 | 0.8017 ± 0.0000 ✓ | 40.55 ± 0.00 | 64.92 ± 0.00 | 0.8065 ± 0.0000 ✓ | 55.71 ± 0.00 | 87.70 ± 0.00 |
| SCALE w/o LF | 0.7973 ± 0.0031 ✓ | 6.54 ± 0.06 | 11.83 ± 0.00 | 0.8009 ± 0.0003 ✓ | 54.02 ± 0.04 | 92.20 ± 0.03 | 0.8159 ± 0.0006 ✓ | 40.68 ± 0.07 | 64.79 ± 0.03 | 0.8243 ± 0.0009 ✗ | 52.76 ± 0.15 | 86.23 ± 0.10 |
| SCALE w/o SGWT | 0.7928 ± 0.0173 ✓ | 6.30 ± 0.29 | 11.15 ± 0.01 | 0.8035 ± 0.0147 ✓ | 53.36 ± 1.86 | 79.39 ± 0.14 | 0.7974 ± 0.0211 ✓ | 39.23 ± 1.94 | 58.79 ± 0.10 | 0.8214 ± 0.0124 ✗ | 54.85 ± 1.45 | 77.40 ± 0.28 |
| SCALE | 0.7877 ± 0.0045 ✓ | 6.37 ± 0.13 | 10.99 ± 0.02 | 0.8077 ± 0.0059 ✓ | **50.19 ± 0.59** | **76.07 ± 0.07** | 0.8157 ± 0.0065 ✓ | **37.35 ± 0.60** | **55.19 ± 0.09** | 0.8028 ± 0.0055 ✓ | **51.54 ± 0.62** | **75.31 ± 0.06** |

*Table 6.* The first stage backbone configuration used to generate residuals.

| Backbone | Source | Key configuration |
|---|---|---|
| GRU + DiffConv template | Our default | See Appendix E |
| Transformer | tsl | Default settings from tsl |
| DCRNN | tsl | Default settings from tsl |
| GWNet | tsl | Default settings from tsl |

*Table 7.* Interval results on METR-LA, PEMS07, PEMS08, PEMS04 at $\alpha \in \{0.05, 0.1, 0.2\}$ for the first stage point forecasting backbone (Transformer). We report Coverage, PI-Width, and Winkler (mean±std over seeds for trainable methods); non-parametric residual-quantile baselines (SCP/SeqCP/NexCP) are deterministic given residuals.

| Method | METR-LA | | | PEMS07 | | | PEMS08 | | | PEMS04 | | |
|---|---|---|---|---|---|---|---|---|---|---|---|---|
| | Coverage ↑ | PI-Width ↓ | Winkler ↓ | Coverage ↑ | PI-Width ↓ | Winkler ↓ | Coverage ↑ | PI-Width ↓ | Winkler ↓ | Coverage ↑ | PI-Width ↓ | Winkler ↓ |
| **$\alpha = 0.05$** | | | | | | | | | | | | |
| SCP | 0.9484 ± 0.0000 ✓ | 14.97 ± 0.00 | 25.00 ± 0.00 | 0.9456 ± 0.0000 ✓ | 101.49 ± 0.00 | 146.60 ± 0.00 | 0.9517 ± 0.0000 ✓ | 76.93 ± 0.00 | 106.78 ± 0.00 | 0.9537 ± 0.0000 ✓ | 107.74 ± 0.00 | 142.83 ± 0.00 |
| SeqCP | 0.9144 ± 0.0000 ✗ | 13.29 ± 0.00 | 25.36 ± 0.00 | 0.9476 ± 0.0000 ✓ | 147.59 ± 0.00 | 186.75 ± 0.00 | 0.9206 ± 0.0000 ✗ | 69.83 ± 0.00 | 110.03 ± 0.00 | 0.9105 ± 0.0000 ✗ | 92.33 ± 0.00 | 150.05 ± 0.00 |
| CoREL | 0.9491 ± 0.0009 ✓ | 13.09 ± 0.17 | **19.07 ± 0.03** | 0.9479 ± 0.0023 ✓ | 88.02 ± 0.58 | 118.97 ± 0.01 | 0.9524 ± 0.0030 ✓ | 65.60 ± 1.43 | 85.80 ± 0.07 | 0.9529 ± 0.0024 ✓ | 87.30 ± 1.13 | **112.07 ± 0.26** |
| NexCP | 0.9438 ± 0.0000 ✓ | 14.68 ± 0.00 | 24.18 ± 0.00 | 0.9463 ± 0.0000 ✓ | 100.51 ± 0.00 | 140.29 ± 0.00 | 0.9461 ± 0.0000 ✓ | 74.60 ± 0.00 | 103.79 ± 0.00 | 0.9441 ± 0.0000 ✓ | 101.52 ± 0.00 | 139.94 ± 0.00 |
| EnbPI | 0.9491 ± 0.0000 ✓ | 14.97 ± 0.00 | 24.77 ± 0.00 | 0.9484 ± 0.0000 ✓ | 100.03 ± 0.01 | 143.18 ± 0.00 | 0.9505 ± 0.0000 ✓ | 75.46 ± 0.02 | 105.40 ± 0.00 | 0.9510 ± 0.0000 ✓ | 104.44 ± 0.01 | 141.23 ± 0.00 |
| ConForME | 0.9488 ± 0.0000 ✓ | 15.26 ± 0.00 | 25.41 ± 0.00 | 0.9461 ± 0.0000 ✓ | 103.90 ± 0.00 | 148.21 ± 0.00 | 0.9518 ± 0.0000 ✓ | 77.11 ± 0.00 | 106.92 ± 0.00 | 0.9543 ± 0.0000 ✓ | 109.28 ± 0.00 | 144.27 ± 0.00 |
| SCALE | 0.9449 ± 0.0020 ✓ | **12.57 ± 0.11** | 19.22 ± 0.05 | 0.9499 ± 0.0028 ✓ | **84.15 ± 0.87** | **118.41 ± 0.04** | 0.9573 ± 0.0016 ✓ | **63.57 ± 0.12** | **83.97 ± 0.28** | 0.9442 ± 0.0025 ✓ | **83.79 ± 0.87** | 113.53 ± 0.16 |
| **$\alpha = 0.10$** | | | | | | | | | | | | |
| SCP | 0.8994 ± 0.0000 ✓ | 10.15 ± 0.00 | 18.70 ± 0.00 | 0.8945 ± 0.0000 ✓ | 78.66 ± 0.00 | 118.08 ± 0.00 | 0.9043 ± 0.0000 ✓ | 59.50 ± 0.00 | 86.05 ± 0.00 | 0.9065 ± 0.0000 ✓ | 84.19 ± 0.00 | 117.13 ± 0.00 |
| SeqCP | 0.8618 ± 0.0000 ✗ | 9.53 ± 0.00 | 18.67 ± 0.00 | 0.8678 ± 0.0000 ✗ | 74.73 ± 0.00 | 118.08 ± 0.00 | 0.8698 ± 0.0000 ✗ | 55.66 ± 0.00 | 87.97 ± 0.00 | 0.8572 ± 0.0000 ✗ | 74.16 ± 0.00 | 120.62 ± 0.00 |
| CoREL | 0.9009 ± 0.0023 ✓ | 9.99 ± 0.24 | **15.24 ± 0.01** | 0.8979 ± 0.0037 ✓ | 71.45 ± 0.43 | 99.31 ± 0.04 | 0.9049 ± 0.0052 ✓ | 53.72 ± 1.09 | 72.10 ± 0.06 | 0.9041 ± 0.0029 ✓ | 71.59 ± 0.72 | **94.79 ± 0.15** |
| NexCP | 0.8945 ± 0.0000 ✓ | 10.01 ± 0.00 | 18.00 ± 0.00 | 0.8966 ± 0.0000 ✓ | 78.30 ± 0.00 | 113.86 ± 0.00 | 0.8972 ± 0.0000 ✓ | 58.10 ± 0.00 | 84.25 ± 0.00 | 0.8943 ± 0.0000 ✓ | 79.45 ± 0.00 | 114.71 ± 0.00 |
| EnbPI | 0.8999 ± 0.0000 ✓ | 10.17 ± 0.00 | 18.56 ± 0.00 | 0.8974 ± 0.0000 ✓ | 77.58 ± 0.01 | 115.75 ± 0.00 | 0.9016 ± 0.0000 ✓ | 58.58 ± 0.01 | 85.31 ± 0.00 | 0.9020 ± 0.0000 ✓ | 81.53 ± 0.01 | 115.85 ± 0.00 |
| ConForME | 0.8997 ± 0.0000 ✓ | 10.13 ± 0.00 | 18.94 ± 0.00 | 0.8948 ± 0.0000 ✓ | 81.34 ± 0.00 | 120.10 ± 0.00 | 0.9048 ± 0.0000 ✓ | 59.67 ± 0.00 | 86.20 ± 0.00 | 0.9067 ± 0.0000 ✓ | 85.53 ± 0.00 | 118.45 ± 0.00 |
| SCALE | 0.8927 ± 0.0031 ✓ | **9.61 ± 0.14** | 15.34 ± 0.02 | 0.9058 ± 0.0050 ✓ | **68.23 ± 0.65** | **97.14 ± 0.09** | 0.9138 ± 0.0022 ✓ | **51.19 ± 0.09** | **69.69 ± 0.25** | 0.8937 ± 0.0045 ✓ | **68.30 ± 0.62** | 95.22 ± 0.09 |
| **$\alpha = 0.20$** | | | | | | | | | | | | |
| SCP | 0.8031 ± 0.0000 ✓ | 6.17 ± 0.00 | 13.25 ± 0.00 | 0.7944 ± 0.0000 ✓ | 56.53 ± 0.00 | 92.41 ± 0.00 | 0.8070 ± 0.0000 ✓ | 42.71 ± 0.00 | 67.48 ± 0.00 | 0.8094 ± 0.0000 ✓ | 59.81 ± 0.00 | 92.66 ± 0.00 |
| SeqCP | 0.7662 ± 0.0000 ✗ | 6.21 ± 0.00 | 13.18 ± 0.00 | 0.7714 ± 0.0000 ✗ | 55.35 ± 0.00 | 92.22 ± 0.00 | 0.7733 ± 0.0000 ✗ | 41.16 ± 0.00 | 68.43 ± 0.00 | 0.7622 ± 0.0000 ✗ | 54.70 ± 0.00 | 94.01 ± 0.00 |
| CoREL | 0.8016 ± 0.0049 ✓ | 7.20 ± 0.18 | **11.83 ± 0.00** | 0.7974 ± 0.0024 ✓ | 54.08 ± 0.05 | 80.81 ± 0.07 | 0.8029 ± 0.0051 ✓ | **40.59 ± 0.58** | 59.07 ± 0.05 | 0.8085 ± 0.0031 ✓ | 54.97 ± 0.43 | 78.11 ± 0.10 |
| NexCP | 0.7946 ± 0.0000 ✓ | 6.18 ± 0.00 | 12.83 ± 0.00 | 0.7974 ± 0.0000 ✓ | 56.78 ± 0.00 | 90.00 ± 0.00 | 0.7979 ± 0.0000 ✓ | 42.08 ± 0.00 | 66.61 ± 0.00 | 0.7947 ± 0.0000 ✓ | 56.85 ± 0.00 | 90.99 ± 0.00 |
| EnbPI | 0.8021 ± 0.0000 ✓ | 6.22 ± 0.00 | 13.21 ± 0.00 | 0.7969 ± 0.0000 ✓ | 55.80 ± 0.00 | 90.82 ± 0.00 | 0.8028 ± 0.0000 ✓ | 42.01 ± 0.01 | 67.03 ± 0.00 | 0.8035 ± 0.0000 ✓ | 57.96 ± 0.01 | 91.61 ± 0.00 |
| ConForME | 0.8033 ± 0.0000 ✓ | **6.12 ± 0.00** | 13.35 ± 0.00 | 0.7948 ± 0.0000 ✓ | 59.16 ± 0.00 | 94.69 ± 0.00 | 0.8073 ± 0.0000 ✓ | 42.76 ± 0.00 | 67.57 ± 0.00 | 0.8100 ± 0.0000 ✓ | 60.89 ± 0.00 | 93.87 ± 0.00 |
| SCALE | 0.7895 ± 0.0071 ✓ | 6.91 ± 0.15 | 11.87 ± 0.02 | 0.8158 ± 0.0075 ✓ | **51.57 ± 0.51** | **78.13 ± 0.12** | 0.8232 ± 0.0057 ✗ | 38.60 ± 0.08 | 56.48 ± 0.21 | 0.7940 ± 0.0065 ✓ | **51.95 ± 0.50** | **77.92 ± 0.05** |

forecasting backbones and often achieves smaller interval widths or Winkler scores than the strongest non-SCALE baselines.

These results support the practical scope of SCALE as a broadly compatible post-hoc calibration method, rather than a method tied to a specific forecasting architecture.

*Table 8.* Interval results on METR-LA, PEMS07, PEMS08, PEMS04 at $\alpha \in \{0.05, 0.1, 0.2\}$ for the first stage point forecasting backbone (GWNet). We report Coverage, PI-Width, and Winkler (mean±std over seeds for trainable methods); non-parametric residual-quantile baselines (SCP/SeqCP/NexCP) are deterministic given residuals.

| Method | METR-LA | | | PEMS07 | | | PEMS08 | | | PEMS04 | | |
|---|---|---|---|---|---|---|---|---|---|---|---|---|
| | Coverage ↑ | PI-Width ↓ | Winkler ↓ | Coverage ↑ | PI-Width ↓ | Winkler ↓ | Coverage ↑ | PI-Width ↓ | Winkler ↓ | Coverage ↑ | PI-Width ↓ | Winkler ↓ |
| $\alpha = 0.05$ | | | | | | | | | | | | |
| SCP | 0.9471±0.0000 ✓ | 13.27±0.00 | 21.95±0.00 | 0.9443±0.0000 ✓ | 89.04±0.00 | 131.04±0.00 | 0.9491±0.0000 ✓ | 69.59±0.00 | 98.61±0.00 | 0.9534±0.0000 ✓ | 99.22±0.00 | 134.41±0.00 |
| SeqCP | 0.9093±0.0000 ✗ | 11.82±0.00 | 21.98±0.00 | 0.9181±0.0000 ✗ | 82.54±0.00 | 131.06±0.00 | 0.9194±0.0000 ✗ | 63.37±0.00 | 100.99±0.00 | 0.9088±0.0000 ✗ | 84.43±0.00 | 140.18±0.00 |
| CoREL | 0.9462±0.0006 ✓ | **11.62±0.05** | 17.92±0.09 | 0.9502±0.0023 ✓ | 81.69±1.36 | **110.10±0.20** | 0.9473±0.0007 ✓ | 61.33±1.05 | 81.77±0.34 | 0.9533±0.0024 ✓ | 83.11±0.79 | **106.90±0.05** |
| NexCP | 0.9437±0.0000 ✓ | 13.11±0.00 | 20.94±0.00 | 0.9449±0.0000 ✓ | 88.68±0.00 | 124.65±0.00 | 0.9452±0.0000 ✓ | 68.09±0.00 | 95.49±0.00 | 0.9442±0.0000 ✓ | 93.24±0.00 | 131.07±0.00 |
| EnbPI | 0.9484±0.0000 ✓ | 13.36±0.00 | 21.73±0.00 | 0.9478±0.0000 ✓ | 88.91±0.00 | 128.29±0.00 | 0.9489±0.0000 ✓ | 68.97±0.02 | 97.63±0.01 | 0.9511±0.0000 ✓ | 97.11±0.01 | 133.58±0.01 |
| ConForME | 0.9473±0.0000 ✓ | 13.57±0.00 | 22.67±0.00 | 0.9447±0.0000 ✓ | 89.86±0.00 | 132.09±0.00 | 0.9488±0.0000 ✓ | 70.38±0.00 | 99.34±0.00 | 0.9538±0.0000 ✓ | 99.55±0.00 | 135.10±0.00 |
| SCALE | 0.9457±0.0023 ✓ | 11.77±0.13 | **17.81±0.05** | 0.9482±0.0025 ✓ | **78.57±1.46** | 110.39±0.27 | 0.9467±0.0046 ✓ | **59.04±1.30** | **80.33±0.11** | 0.9476±0.0040 ✓ | **82.10±1.46** | 108.82±0.11 |
| $\alpha = 0.10$ | | | | | | | | | | | | |
| SCP | 0.8972±0.0000 ✓ | 9.21±0.00 | 16.61±0.00 | 0.8928±0.0000 ✓ | 69.12±0.00 | 105.16±0.00 | 0.8998±0.0000 ✓ | 53.96±0.00 | 79.26±0.00 | 0.9059±0.0000 ✓ | 76.89±0.00 | 109.11±0.00 |
| SeqCP | 0.8552±0.0000 ✗ | 8.59±0.00 | 16.40±0.00 | 0.8649±0.0000 ✗ | 65.75±0.00 | 104.52±0.00 | 0.8689±0.0000 ✗ | 50.58±0.00 | 80.33±0.00 | 0.8553±0.0000 ✗ | 67.58±0.00 | 111.75±0.00 |
| CoREL | 0.8962±0.0022 ✓ | **8.67±0.14** | **14.08±0.04** | 0.9004±0.0031 ✓ | 66.03±0.91 | 91.58±0.05 | 0.8949±0.0022 ✓ | 49.80±0.77 | 68.64±0.27 | 0.9053±0.0020 ✓ | 68.10±0.39 | **90.19±0.05** |
| NexCP | 0.8934±0.0000 ✓ | 9.11±0.00 | 15.83±0.00 | 0.8950±0.0000 ✓ | 69.02±0.00 | 100.89±0.00 | 0.8962±0.0000 ✓ | 53.00±0.00 | 77.30±0.00 | 0.8936±0.0000 ✓ | 72.48±0.00 | 106.44±0.00 |
| EnbPI | 0.8987±0.0000 ✓ | 9.23±0.00 | 16.47±0.00 | 0.8967±0.0000 ✓ | 68.92±0.00 | 103.53±0.00 | 0.8990±0.0001 ✓ | 53.48±0.00 | 78.80±0.00 | 0.9025±0.0000 ✓ | 75.46±0.01 | 108.64±0.00 |
| ConForME | 0.8972±0.0000 ✓ | 9.06±0.00 | 16.96±0.00 | 0.8940±0.0000 ✓ | 70.28±0.00 | 106.10±0.00 | 0.8998±0.0000 ✓ | 54.57±0.00 | 80.03±0.00 | 0.9063±0.0000 ✓ | 77.11±0.00 | 109.54±0.00 |
| SCALE | 0.8943±0.0030 ✓ | 8.87±0.06 | 14.15±0.03 | 0.9013±0.0032 ✓ | **63.78±1.15** | **90.47±0.12** | 0.8960±0.0083 ✓ | **47.72±1.14** | **66.69±0.08** | 0.8995±0.0050 ✓ | **67.07±1.15** | 91.03±0.11 |
| $\alpha = 0.20$ | | | | | | | | | | | | |
| SCP | 0.7962±0.0000 ✓ | 5.69±0.00 | 11.94±0.00 | 0.7926±0.0000 ✓ | 49.90±0.00 | 82.14±0.00 | 0.8002±0.0000 ✓ | 38.76±0.00 | 62.03±0.00 | 0.8080±0.0000 ✓ | 54.20±0.00 | 85.58±0.00 |
| SeqCP | 0.7568±0.0000 ✗ | 5.70±0.00 | 11.75±0.00 | 0.7663±0.0000 ✗ | 48.74±0.00 | 81.48±0.00 | 0.7740±0.0000 ✗ | 37.59±0.00 | 62.51±0.00 | 0.7609±0.0000 ✗ | 49.69±0.00 | 86.45±0.00 |
| CoREL | 0.7993±0.0049 ✓ | 6.29±0.11 | **10.78±0.01** | 0.8028±0.0037 ✓ | 49.99±0.43 | 74.29±0.02 | 0.7907±0.0045 ✓ | 37.74±0.41 | 56.16±0.21 | 0.8089±0.0024 ✓ | 52.18±0.38 | **74.22±0.04** |
| NexCP | 0.7923±0.0000 ✓ | 5.77±0.00 | 11.45±0.00 | 0.7957±0.0000 ✓ | 50.13±0.00 | 79.64±0.00 | 0.7974±0.0000 ✓ | 38.47±0.00 | 61.00±0.00 | 0.7936±0.0000 ✓ | 51.56±0.00 | 83.76±0.00 |
| EnbPI | 0.7985±0.0000 ✓ | 5.69±0.00 | 11.85±0.00 | 0.7957±0.0000 ✓ | 49.56±0.00 | 81.11±0.00 | 0.7987±0.0001 ✓ | 38.42±0.00 | 61.82±0.00 | 0.8032±0.0000 ✓ | 53.24±0.01 | 85.36±0.00 |
| ConForME | 0.7967±0.0000 ✓ | **5.59±0.00** | 12.05±0.00 | 0.7946±0.0000 ✓ | 51.21±0.00 | 83.10±0.00 | 0.8001±0.0000 ✓ | 39.22±0.00 | 62.72±0.00 | 0.8084±0.0000 ✓ | 54.40±0.00 | 85.86±0.00 |
| SCALE | 0.7928±0.0041 ✓ | 6.28±0.04 | 10.85±0.01 | 0.8062±0.0045 ✓ | **48.32±0.81** | **72.75±0.05** | 0.7948±0.0125 ✓ | **36.01±0.89** | **54.11±0.07** | 0.8006±0.0095 ✓ | **51.13±0.96** | 74.39±0.12 |

*Table 9.* Interval results on METR-LA, PEMS07, PEMS08, PEMS04 at $\alpha \in \{0.05, 0.1, 0.2\}$ for the first stage point forecasting backbone (DCRNN). We report Coverage, PI-Width, and Winkler (mean±std over seeds for trainable methods); non-parametric residual-quantile baselines (SCP/SeqCP/NexCP) are deterministic given residuals.

| Method | METR-LA | | | PEMS07 | | | PEMS08 | | | PEMS04 | | |
|---|---|---|---|---|---|---|---|---|---|---|---|---|
| | Coverage ↑ | PI-Width ↓ | Winkler ↓ | Coverage ↑ | PI-Width ↓ | Winkler ↓ | Coverage ↑ | PI-Width ↓ | Winkler ↓ | Coverage ↑ | PI-Width ↓ | Winkler ↓ |
| $\alpha = 0.05$ | | | | | | | | | | | | |
| SCP | 0.9480±0.0000 ✓ | 13.81±0.00 | 22.41±0.00 | 0.9459±0.0000 ✓ | 95.91±0.00 | 139.18±0.00 | 0.9493±0.0000 ✓ | 75.15±0.00 | 106.37±0.00 | 0.9531±0.0000 ✓ | 103.49±0.00 | 138.70±0.00 |
| SeqCP | 0.9129±0.0000 ✗ | 12.28±0.00 | 22.72±0.00 | 0.9209±0.0000 ✗ | 88.75±0.00 | 139.98±0.00 | 0.9183±0.0000 ✗ | 68.58±0.00 | 108.90±0.00 | 0.9097±0.0000 ✗ | 88.67±0.00 | 145.17±0.00 |
| CoREL | 0.9503±0.0035 ✓ | 12.29±0.15 | **18.24±0.03** | 0.9475±0.0038 ✓ | 85.87±0.12 | **115.41±0.19** | 0.9482±0.0037 ✓ | 65.12±1.28 | 86.17±0.29 | 0.9537±0.0020 ✓ | 86.22±0.82 | **110.53±0.20** |
| NexCP | 0.9440±0.0000 ✓ | 13.58±0.00 | 21.66±0.00 | 0.9457±0.0000 ✓ | 95.04±0.00 | 133.20±0.00 | 0.9459±0.0000 ✓ | 73.70±0.00 | 102.36±0.00 | 0.9441±0.0000 ✓ | 97.72±0.00 | 135.43±0.00 |
| EnbPI | 0.9487±0.0000 ✓ | 13.88±0.00 | 22.32±0.00 | 0.9485±0.0000 ✓ | 95.60±0.01 | 136.66±0.00 | 0.9501±0.0001 ✓ | 73.89±0.03 | 104.24±0.01 | 0.9508±0.0001 ✓ | 100.96±0.01 | 137.58±0.00 |
| ConForME | 0.9485±0.0000 ✓ | 14.07±0.00 | 22.98±0.00 | 0.9461±0.0000 ✓ | 96.14±0.00 | 139.67±0.00 | 0.9496±0.0000 ✓ | 76.25±0.00 | 107.66±0.00 | 0.9535±0.0000 ✓ | 104.01±0.00 | 139.48±0.00 |
| SCALE | 0.9444±0.0030 ✓ | **12.09±0.25** | 18.32±0.08 | 0.9443±0.0043 ✓ | **81.68±0.87** | 115.79±0.30 | 0.9543±0.0018 ✓ | **63.28±0.46** | **83.77±0.17** | 0.9425±0.0047 ✓ | **83.06±1.81** | 111.89±0.13 |
| $\alpha = 0.10$ | | | | | | | | | | | | |
| SCP | 0.8984±0.0000 ✓ | 9.57±0.00 | 17.01±0.00 | 0.8948±0.0000 ✓ | 74.28±0.00 | 111.87±0.00 | 0.9010±0.0000 ✓ | 58.16±0.00 | 85.47±0.00 | 0.9057±0.0000 ✓ | 80.70±0.00 | 113.34±0.00 |
| SeqCP | 0.8590±0.0000 ✗ | 8.96±0.00 | 16.98±0.00 | 0.8692±0.0000 ✗ | 70.66±0.00 | 111.77±0.00 | 0.8679±0.0000 ✗ | 54.80±0.00 | 87.09±0.00 | 0.8562±0.0000 ✗ | 71.12±0.00 | 116.35±0.00 |
| CoREL | 0.9034±0.0044 ✓ | 9.31±0.08 | **14.45±0.02** | 0.8967±0.0053 ✓ | 69.56±0.10 | 96.32±0.13 | 0.8961±0.0048 ✓ | 52.68±0.82 | 72.45±0.23 | 0.9055±0.0042 ✓ | 70.68±0.83 | **93.40±0.13** |
| NexCP | 0.8945±0.0000 ✓ | 9.46±0.00 | 16.39±0.00 | 0.8963±0.0000 ✓ | 73.89±0.00 | 107.88±0.00 | 0.8969±0.0000 ✓ | 57.34±0.00 | 83.17±0.00 | 0.8938±0.0000 ✓ | 76.25±0.00 | 110.65±0.00 |
| EnbPI | 0.8991±0.0000 ✓ | 9.60±0.00 | 16.96±0.00 | 0.8978±0.0000 ✓ | 74.12±0.00 | 110.43±0.00 | 0.9007±0.0000 ✓ | 57.24±0.01 | 84.10±0.01 | 0.9018±0.0001 ✓ | 78.82±0.01 | 112.54±0.01 |
| ConForME | 0.8985±0.0000 ✓ | 9.46±0.00 | 17.30±0.00 | 0.8951±0.0000 ✓ | 74.53±0.00 | 112.20±0.00 | 0.9007±0.0000 ✓ | 59.25±0.00 | 86.69±0.00 | 0.9061±0.0000 ✓ | 81.09±0.00 | 113.90±0.00 |
| SCALE | 0.8923±0.0030 ✓ | **9.16±0.15** | 14.58±0.02 | 0.8958±0.0067 ✓ | **66.32±0.76** | **95.19±0.19** | 0.9076±0.0029 ✓ | **51.21±0.32** | **69.91±0.10** | 0.8908±0.0060 ✓ | **67.70±1.39** | 93.89±0.07 |
| $\alpha = 0.20$ | | | | | | | | | | | | |
| SCP | 0.7989±0.0000 ✓ | 5.94±0.00 | 12.27±0.00 | 0.7943±0.0000 ✓ | 53.43±0.00 | 87.43±0.00 | 0.8021±0.0000 ✓ | 41.68±0.00 | 66.88±0.00 | 0.8080±0.0000 ✓ | 57.07±0.00 | 89.35±0.00 |
| SeqCP | 0.7621±0.0000 ✗ | 5.96±0.00 | 12.19±0.00 | 0.7736±0.0000 ✗ | 52.23±0.00 | 87.07±0.00 | 0.7730±0.0000 ✗ | 40.49±0.00 | 67.68±0.00 | 0.7617±0.0000 ✗ | 52.46±0.00 | 90.51±0.00 |
| CoREL | 0.8099±0.0050 ✓ | 6.79±0.04 | **11.14±0.01** | 0.7967±0.0083 ✓ | 52.77±0.13 | 78.38±0.07 | 0.7944±0.0024 ✓ | 40.06±0.25 | 59.40±0.18 | 0.8106±0.0050 ✓ | 54.21±0.48 | 76.95±0.07 |
| NexCP | 0.7938±0.0000 ✓ | 6.00±0.00 | 11.87±0.00 | 0.7970±0.0000 ✓ | 53.51±0.00 | 85.12±0.00 | 0.7982±0.0000 ✓ | 41.48±0.00 | 65.71±0.00 | 0.7937±0.0000 ✓ | 54.44±0.00 | 87.55±0.00 |
| EnbPI | 0.7993±0.0000 ✓ | 5.97±0.00 | 12.25±0.00 | 0.7973±0.0000 ✓ | 53.29±0.00 | 86.63±0.00 | 0.8014±0.0000 ✓ | 41.06±0.01 | 65.97±0.01 | 0.8028±0.0000 ✓ | 55.81±0.00 | 88.82±0.00 |
| ConForME | 0.7993±0.0000 ✓ | **5.85±0.00** | 12.36±0.00 | 0.7948±0.0000 ✓ | 53.59±0.00 | 87.63±0.00 | 0.8018±0.0000 ✓ | 42.80±0.00 | 68.07±0.00 | 0.8080±0.0000 ✓ | 57.32±0.00 | 89.78±0.00 |
| SCALE | 0.7905±0.0026 ✓ | 6.53±0.08 | 11.23±0.01 | 0.7989±0.0109 ✓ | **50.20±0.58** | **76.66±0.12** | 0.8116±0.0043 ✓ | **38.59±0.30** | **56.84±0.10** | 0.7901±0.0090 ✓ | **51.69±1.09** | **76.87±0.01** |

*Table 10.* Additional interval results with stronger first-stage forecasting backbones at $\alpha = 0.1$. Cov., Width, and Winkler denote empirical coverage, prediction interval width, and Winkler score, respectively.

| Dataset | Metric | SCP | SeqCP | NexCP | EnbPI | ConForME | CoREL | SCALE |
|---|---|---|---|---|---|---|---|---|
| **PatchSTG (Fang et al., 2024)** | | | | | | | | |
| PEMS08 | Cov. | 0.9027 | 0.8690 | 0.8967 | 0.9010 | 0.9034 | 0.9005 | 0.9072 |
| | Width | 52.8123 | 49.2458 | 51.5682 | 52.2733 | 53.0917 | 48.0826 | **47.6730** |
| | Winkler | 77.4113 | 78.1120 | 75.1661 | 76.8026 | 77.7820 | 66.0298 | **65.9557** |
| AirQuality | Cov. | 0.9307 | 0.8781 | 0.9015 | 0.9336 | 0.9430 | 0.9098 | 0.9095 |
| | Width | 67.2491 | **39.7247** | 42.1153 | 56.3539 | 68.3677 | 42.7885 | 41.5345 |
| | Winkler | 92.7544 | 75.4840 | 73.6517 | 82.4830 | 93.0697 | 65.8700 | **65.3689** |
| Electricity | Cov. | 0.8906 | 0.8832 | 0.8952 | 0.8890 | 0.8947 | 0.8910 | 0.8966 |
| | Width | 687.8394 | 680.1237 | 710.4052 | 639.6601 | 697.9498 | 549.2029 | **504.2981** |
| | Winkler | 1120.9625 | 1086.6011 | 1088.3804 | 1068.8358 | 1158.2959 | 957.6906 | **955.7571** |
| USHCN-West | Cov. | 0.8938 | 0.8774 | 0.8948 | 0.8941 | 0.8949 | 0.8978 | 0.8762 |
| | Width | 11.4567 | 11.0672 | 11.5402 | 11.3730 | 11.5505 | 11.2895 | **10.6484** |
| | Winkler | 16.1780 | 16.0670 | 15.8543 | 16.0081 | 16.2591 | **15.4855** | 15.8081 |
| **MAGE (Ma et al., 2025)** | | | | | | | | |
| PEMS08 | Cov. | 0.8967 | 0.8683 | 0.8961 | 0.8984 | 0.8966 | 0.9079 | 0.8964 |
| | Width | 52.4797 | 49.2739 | 51.7584 | 51.9189 | 53.0375 | 51.1141 | **48.0223** |
| | Winkler | 77.5986 | 77.7718 | 74.8555 | 76.4647 | 78.3303 | 66.8541 | **66.4269** |
| AirQuality | Cov. | 0.9474 | 0.8646 | 0.8995 | 0.9317 | 0.9314 | 0.9173 | 0.9229 |
| | Width | 87.1263 | 55.7790 | 60.6942 | 62.0561 | 100.2566 | 53.3899 | **52.0678** |
| | Winkler | 118.0309 | 96.3624 | 93.6856 | 91.5909 | 139.1659 | 76.9869 | **75.3734** |
| Electricity | Cov. | 0.8828 | 0.8819 | 0.8952 | 0.8843 | 0.8859 | 0.8879 | 0.8737 |
| | Width | 675.9958 | 671.1943 | 701.6153 | 626.2939 | 744.9166 | 531.9865 | **492.1814** |
| | Winkler | 1112.1085 | 1054.1091 | 1056.8157 | 1049.6592 | 1207.3130 | 952.5536 | **915.6543** |
| USHCN-West | Cov. | 0.8918 | 0.8777 | 0.8948 | 0.8933 | 0.8921 | 0.8988 | 0.8894 |
| | Width | 11.3579 | 11.0895 | 11.5351 | 11.3319 | 11.3927 | 11.2895 | **10.9971** |
| | Winkler | 16.1545 | 16.0769 | 15.8645 | 16.0201 | 16.2156 | 15.4863 | **15.4133** |
| **D2STGNN (Shao et al., 2022b)** | | | | | | | | |
| PEMS08 | Cov. | 0.9031 | 0.8576 | 0.8959 | 0.9009 | 0.9024 | 0.9060 | 0.8989 |
| | Width | 51.1232 | 47.4813 | 49.9935 | 50.0620 | 51.8980 | 48.7626 | **45.7966** |
| | Winkler | 74.8285 | 75.3450 | 72.5213 | 73.8160 | 75.7415 | 64.3873 | **63.9071** |
| AirQuality | Cov. | 0.9315 | 0.8766 | 0.9001 | 0.9329 | 0.9413 | 0.9254 | 0.9031 |
| | Width | 64.5568 | 41.7032 | 43.9708 | 56.9561 | 65.3994 | 45.7076 | **40.7523** |
| | Winkler | 95.2047 | 79.8397 | 77.8813 | 85.8925 | 95.7189 | 66.5374 | **66.5153** |
| Electricity | Cov. | 0.8899 | 0.8833 | 0.8951 | 0.8867 | 0.8907 | 0.9114 | 0.9129 |
| | Width | 690.9509 | 729.3012 | 766.0544 | 673.0826 | 708.3678 | **592.2473** | 596.6196 |
| | Winkler | 1273.0098 | 1226.5300 | 1235.1464 | 1252.3036 | 1309.2285 | 1056.2280 | **1036.3817** |
| USHCN-West | Cov. | 0.8921 | 0.8775 | 0.8950 | 0.8932 | 0.8932 | 0.9007 | 0.8985 |
| | Width | 11.3315 | **11.0421** | 11.5023 | 11.3056 | 11.4165 | 11.4184 | 11.3883 |
| | Winkler | 16.1430 | 16.0574 | 15.8562 | 16.0260 | 16.2610 | 15.5124 | **15.3741** |

