# OpenReview forum: "Delving into Non-Exchangeability for Conformal Prediction in Graph-Structured Multivariate Time Series"
_ICML.cc/2026/Conference — ICML 2026 regular_

### Official Review · Reviewer_CA3r · 2026-02-15

**Soundness:** 3
**Presentation:** 3
**Significance:** 2
**Originality:** 3
**Overall Recommendation:** 3
**Confidence:** 4

**Summary:**

This paper focuses on point prediction for graph-structured multivariate time series.
It introduces a novel concept termed "Spectral Graph Conditional Exchangeability" (SGCE),
which conditionally exchanges high-frequency components given low-frequency components to
preserve global trends and enable effective conformal prediction (CP) in the spectral domain.
Experimental results on real-world traffic datasets demonstrate that SCALE consistently
outperforms state-of-the-art CP methods in balancing coverage efficiency and coverage rate.

**Compliance With Llm Reviewing Policy:**

Affirmed.

**Key Questions For Authors:**

Please refer to the weakness.

**Limitations:**

Yes.

**Strengths And Weaknesses:**

> Strengths

The paper addresses spatiotemporal point process forecasting, which is a novel and underexplored research problem.

The figures and diagrams are thoughtfully designed and enhance the clarity of the presentation.

The proposed model appears to be well-motivated and reasonably structured.

> Questions for the Authors

Q1. Could the authors clarify the fundamental distinction between point process forecasting and conventional spatiotemporal forecasting? Are the datasets used identical to those commonly adopted in standard spatiotemporal prediction tasks? If so, what unique challenges does the point process formulation introduce? Given that this problem appears—based on its definition—to be potentially equivalent to conventional spatiotemporal forecasting, a thorough justification of its novelty and distinct challenges would be essential for readers to appreciate the contribution.

Q2. The term "non-exchangeability challenge" is central to the paper but lacks a clear definition. Could the authors provide a precise explanation of this concept and, ideally, include an illustrative figure to enhance understanding? Additionally, the sentence "To this end, conformal prediction commonly assumes exchangeability, which means that the joint distribution of examples in a held-out calibration set is invariant under the permutation of their indices (Angelopoulos & Bates, 2021)" appears fragmented and conceptually dense—could the authors elaborate on how the assumption of exchangeability in conformal prediction conflicts with the non-exchangeability inherent in spatiotemporal point processes?

Q3. The experimental evaluation relies solely on a simple backbone architecture ("GRU + DiffConv"). Since the proposed framework appears to be general-purpose, could the authors demonstrate its compatibility and effectiveness when integrated with more advanced spatiotemporal backbones (e.g., D2STGNN, MAGE, PatchTST) or specialized point process models? Furthermore, the set of baseline methods appears limited in scope and recency—could the authors enrich the comparison by including recent point process forecasting methods, state-of-the-art spatiotemporal predictors, and diffusion-based uncertainty quantification models to provide a more comprehensive evaluation?

Q4. Several established point process prediction methods—such as DSTPP—are absent from the comparison. Could the authors include these relevant baselines to enable a more rigorous and fair assessment of the proposed model's effectiveness in the spatiotemporal point process forecasting setting?

Q5. The paper lacks details on data preprocessing methodology. Specifically, how are conventional spatiotemporal forecasting datasets transformed into spatiotemporal point process forecasting datasets? Could the authors explicitly describe this conversion procedure, including event discretization strategies, temporal aggregation rules, or any assumptions made in constructing the point process representation?

Q6. Could the authors clarify the precise definitions and interpretations of the three evaluation metrics reported in the experiments? Understanding what each metric quantifies—particularly in the context of point process forecasting—would help readers properly assess the reported results.

---

> ### Author Rebuttal · Authors · 2026-03-31
>
> **Re: “The distinction between the task studied in this paper and standard spatiotemporal forecasting is unclear.”**
>
> Thank you for the comments. We believe there may be a misunderstanding about the task studied in our paper. Our paper is **not** about spatiotemporal point-process prediction. Instead, we study the problem of **post-hoc conformal uncertainty quantification for graph-structured multivariate time-series**. In this task, **the purpose of conformal prediction is to provide uncertainty quantification for a given forecast, so that downstream users know not only the prediction itself, but also how reliable it is**. Concretely, a pretrained forecasting backbone first outputs a **point forecast** (a single predicted value), and then our method constructs a **prediction interval** with respect to this forecast. The datasets are used directly in their original time-series form and are **not** converted to event sequences.
>
>
> **Re: “The non-exchangeability challenge is not clearly defined.”**
>
> We clarify the notion of non-exchangeability in our setting as follows. In standard conformal prediction, the calibration samples are assumed to be **exchangeable**, meaning that their joint distribution does not change if we reorder them. In our setting, however, each calibration sample takes the form of a **whole-graph residual snapshot at a single time step** rather than an isolated scalar. Because nodes are spatially coupled and influenced by shared global dynamics, these graph-level residual snapshots are jointly dependent, so reordering them does not in general preserve the joint distribution. This is what we mean by **non-exchangeability** in our paper. The difficulty is that standard conformal prediction relies on exchangeability to justify its finite-sample coverage guarantee; once this condition is violated, the standard coverage guarantee no longer holds in general, and directly applying conformal calibration to the original graph residuals can therefore lead to unreliable prediction intervals.
>
>
> **Re: “Only one simple backbone is used; can the method work with stronger backbones?”**
> This point is already partially addressed in Appendix E.1, where we evaluate SCALE with multiple forecasting backbones, including Transformer, GWNet, and DCRNN. These results show that SCALE is a general **post-hoc calibration method** and is not tied to one specific backbone of the predictor. We will make this clearer in the revision.
>
> **Re: “Why are point-process baselines such as DSTPP not included?”**
>
> We want to clarify that the point-process methods (such as DSTPP) address a different task. DSTPP and related methods are designed for **event-sequence / point-process modeling**, whereas our paper studies the problem of **prediction interval estimation for graph-structured time series**. Our baseline methods are therefore chosen from the appropriate comparison space: uncertainty-quantification and interval-prediction methods for time-series forecasting.
>
> **Re: “How are standard spatiotemporal datasets converted into point-process data?”**
>
> Actually, they are not converted in our method, and we directly use the standard graph time-series benchmarks in their original form.
>
> **Re: “Please clarify the evaluation metrics.”**
>
> We will make the evaluation metrics clearer in the revised manuscript. Suppose the method outputs an interval $[L_t, U_t]$ for the true target $y_t$ . **Coverage** checks whether $y_t$ falls inside the interval, so higher coverage means the interval is more often correct. **PI-Width** is simply $U_t - L_t$ , so it measures how wide the interval is: a larger value means the prediction is less precise and vice versa. However, a method can always get high coverage by making the interval excessively wide. **Winkler** is used to balance these two aspects: it rewards narrow intervals when $y_t \in [L_t, U_t]$ , but adds an extra penalty when $y_t$ falls outside the interval. The ideal method should achieve **high Coverage**, **small PI-Width**, and thus a **low Winkler score**, which indicates a better coverage-efficiency trade-off.  In summary, these metrics provide a comprehensive evaluation of an interval prediction method from multiple perspectives.

---

> > ### Author Rebuttal · Reviewer_CA3r · 2026-04-03
> >
> > The rebuttal does not directly address my concern regarding the claimed generality of the method. The paper presents the proposed approach as highly general and even implies that it can be applied effectively and consistently across a wide range of models. However, the current experimental evaluation remains limited to a few relatively simple baseline models that are no longer at the forefront of the field. It still lacks sufficient validation with more complex and representative models, such as D2STGNN, MAGE, and PatchSTG. Therefore, the existing empirical evidence is not strong enough to support the claim of universal effectiveness. Moreover, the reported performance gains are themselves rather modest, making it difficult to disentangle whether these improvements genuinely come from the proposed method or instead result from increased parameter count, training randomness, or additional hyperparameter tuning. For these reasons, I believe the paper somewhat overstates the effectiveness of the method.

---

> > > ### Author Response · Authors · 2026-04-05
> > >
> > > Thank you for continuing to review our rebuttal and for further raising the issue of whether the claimed generality of the method is adequately supported by the evidence.
> > >
> > > More precisely, SCALE is a **post-hoc conformal calibration framework** whose second stage is decoupled from the first-stage forecaster, making it broadly compatible with different backbones. Its calibration acts on the residual representation and its exchangeability improvement, rather than on the architectural design of the forecasting model itself.
> > >
> > > To strengthen the empirical support, we further evaluate SCALE on stronger and more recent backbones, namely **PatchSTG**, **MAGE**, and **D2STGNN**, across both traffic and non-traffic datasets at confidence level $\alpha = 0.1$. The results are given below where Cov/W/Wink denote Coverage/Width/Winkler.
> > >
> > > |Backbone|Dataset|Metric|SCP|SeqCP|NexCP|EnbPI|ConForME|CoReL|SCALE|
> > > |---|---|---|---:|---:|---:|---:|---:|---:|---:|
> > > |**PatchSTG**|PEMS08|Cov|0.9027|0.8690|0.8967|0.9010|0.9034|0.9005|0.9072|
> > > |||W|52.8123|49.2458|51.5682|52.2733|53.0917|48.0826|**47.6730**|
> > > |||Wink|77.4113|78.1120|75.1661|76.8026|77.7820|66.0298|**65.9557**|
> > > ||AirQuality|Cov|0.9307|0.8781|0.9015|0.9336|0.9430|0.9098|0.9095|
> > > |||W|67.2491|**39.7247**|42.1153|56.3539|68.3677|42.7885|41.5345|
> > > |||Wink|92.7544|75.4840|73.6517|82.4830|93.0697|65.8700|**65.3689**|
> > > ||Electricity|Cov|0.8906|0.8832|0.8952|0.8890|0.8947|0.8910|0.8966|
> > > |||W|687.8394|680.1237|710.4052|639.6601|697.9498|549.2029|**504.2981**|
> > > |||Wink|1120.9625|1086.6011|1088.3804|1068.8358|1158.2959|957.6906|**955.7571**|
> > > ||Ushcn_West|Cov|0.8938|0.8774|0.8948|0.8941|0.8949|0.8978|0.8762|
> > > |||W|11.4567|11.0672|11.5402|11.3730|11.5505|11.2895|**10.6484**|
> > > |||Wink|16.1780|16.0670|15.8543|16.0081|16.2591|**15.4855**|15.8081|
> > > |**MAGE**|PEMS08|Cov|0.8967|0.8683|0.8961|0.8984|0.8966|0.9079|0.8964|
> > > |||W|52.4797|49.2739|51.7584|51.9189|53.0375|51.1141|**48.0223**|
> > > |||Wink|77.5986|77.7718|74.8555|76.4647|78.3303|66.8541|**66.4269**|
> > > ||AirQuality|Cov|0.9474|0.8646|0.8995|0.9317|0.9314|0.9173|0.9229|
> > > |||W|87.1263|55.7790|60.6942|62.0561|100.2566|53.3899|**52.0678**|
> > > |||Wink|118.0309|96.3624|93.6856|91.5909|139.1659|76.9869|**75.3734**|
> > > ||Electricity|Cov|0.8828|0.8819|0.8952|0.8843|0.8859|0.8879|0.8737|
> > > |||W|675.9958|671.1943|701.6153|626.2939|744.9166|531.9865|**492.1814**|
> > > |||Wink|1112.1085|1054.1091|1056.8157|1049.6592|1207.3130|952.5536|**915.6543**|
> > > ||Ushcn_West|Cov|0.8918|0.8777|0.8948|0.8933|0.8921|0.8988|0.8894|
> > > |||W|11.3579|11.0895|11.5351|11.3319|11.3927|11.2895|**10.9971**|
> > > |||Wink|16.1545|16.0769|15.8645|16.0201|16.2156|15.4863|**15.4133**|
> > > |**D2STGNN**|PEMS08|Cov|0.9031|0.8576|0.8959|0.9009|0.9024|0.9060|0.8989|
> > > |||W|51.1232|47.4813|49.9935|50.0620|51.8980|48.7626|**45.7966**|
> > > |||Wink|74.8285|75.3450|72.5213|73.8160|75.7415|64.3873|**63.9071**|
> > > ||AirQuality|Cov|0.9315|0.8766|0.9001|0.9329|0.9413|0.9254|0.9031|
> > > |||W|64.5568|41.7032|43.9708|56.9561|65.3994|45.7076|**40.7523**|
> > > |||Wink|95.2047|79.8397|77.8813|85.8925|95.7189|66.5374|**66.5153**|
> > > ||Electricity|Cov|0.8899|0.8833|0.8951|0.8867|0.8907|0.9114|0.9129|
> > > |||W|690.9509|729.3012|766.0544|673.0826|708.3678|**592.2473**|596.6196|
> > > |||Wink|1273.0098|1226.5300|1235.1464|1252.3036|1309.2285|1056.2280|**1036.3817**|
> > > ||Ushcn_West|Cov|0.8921|0.8775|0.8950|0.8932|0.8932|0.9007|0.8985|
> > > |||W|11.3315|**11.0421**|11.5023|11.3056|11.4165|11.4184|11.3883|
> > > |||Wink|16.1430|16.0574|15.8562|16.0260|16.2610|15.5124|**15.3741**|
> > >
> > > These results further support the broad compatibility of SCALE with stronger modern forecasting backbones. The gains are not uniformly marginal; in several settings, SCALE reduces interval width by around **4%–6%** even over the strongest non-SCALE baseline. Relative to the **second-best baseline**, the gains are larger and in some cases exceed **10%**, which is difficult to attribute to random variation alone. In the final version, we will state this claim more precisely as **broad compatibility across forecasting backbones**, rather than as uniform effectiveness across models.
> > >
> > > Regarding the concern that the observed gains are due to increased complexity, training randomness, or additional tuning, we refer the reviewer to our response to **Reviewer DHCu** for the runtime analysis, and note that the paper already specifies the shared evaluation setup (**Sec. 6.1**), the use of **five random seeds** (**Appendix D.5**), and the full optimization settings and method-specific hyperparameters (**Appendix D.6**). Moreover, **Sec. 6.4** shows that SCALE remains close to the nominal coverage level with only mild variation in efficiency across a broad range of hyperparameter settings.
> > >
> > > Overall, we appreciate the concern about possible overclaiming. Based on the added results on stronger backbones and the architectural decoupling of SCALE from the first-stage forecaster, we believe the transferability claim is supported. In the final version, we will further tighten the wording to state its practical scope more precisely.

---

### Official Review · Reviewer_DHCu · 2026-02-17

**Soundness:** 2
**Presentation:** 2
**Significance:** 2
**Originality:** 2
**Overall Recommendation:** 3
**Confidence:** 3

**Summary:**

This paper develop a graph-wavelet conformal prediction framework, SGCE, that adapts conformal quantiles to calibrate high-frequency components conditioned on low-frequency ones and demonstrate an improved coverage efficiency trade-off on real-world traffic datasets.

**Compliance With Llm Reviewing Policy:**

Affirmed.

**Final Justification:**

Thank you for the rebuttal. The authors clarified some of my concerns, so I am raising my score from 2 to 3.

However, after reading the rebuttal together with the other reviews, I believe the paper would require substantial revision. The issues in writing and organization make it difficult to fully understand the core contribution and to assess whether the empirical results adequately support the claims. These issues are not likely to be resolved by small changes or additional explanation in the current version.

**Key Questions For Authors:**

Please refer to the weaknesses.

**Limitations:**

Please refer to the weaknesses.

**Strengths And Weaknesses:**

Strengths
1. The paper formalizes non-exchangeability in graph multivariate time series through an explicit hypothesis (SGCE) and derives finite-sample coverage guarantees under that assumption.
2. On four traffic benchmarks, the reported results suggest near-nominal coverage with narrower intervals.

Weaknesses
1. The paper under-discusses related work: spectral-domain graph signals filtering and graph-wavelet multi-scale methods are already common in spatial-tempotal traffic and multivariate forecasting, but the paper does not clearly state what is new or provide direct comparisons to these lines of work.

2. The description of the “conformal/calibration” part (Section 3.2) is not sufficiently explicit. It is not clear, at inference time, what exact score is calibrated, what data split is used, what quantile is computed, and where that correction is applied to produce the final interval. As written, the method can be interpreted as training a quantile/interval network with extra training constraints, which is not the same as a clearly specified split-conformal procedure.

3. The paper largely argues by intuition (for example, “the irregular part looks less coupled”), but it does not show that the key condition needed by the guarantee is even approximately true on the datasets where gains are reported.

4. Empirically, the evaluation is too narrow to support the scope of the claims. Results are limited to a small set of highly similar traffic datasets with similar graph construction and similar dynamics. There is no clear stress test where the paper deliberately breaks the assumptions (strong local coupling in the irregular part, different graph types, different domains) to show the boundary of validity.

5. The theory is largely conditional and the approximate guarantee is not operational. The main result holds only under a strong assumption（Theorem 5.1）, and the relaxation (Theorem 5.2 / Appendix C)  introduces terms (δ_leak, δ_dep, Lipschitz constants) . However, the paper does not provide a practical procedure or empirical estimates for these quantities, so the bound offers limited guidance on when and why coverage may fail.

6. The method appears to require expensive spectral computations and a data-driven cutoff selection, but the author does not clearly state the computational cost.

---

> ### Author Rebuttal · Authors · 2026-03-31
>
> **Re: “The relation to prior spectral graph filtering / wavelet work is unclear.”**
>
> We clarify that our paper does **not** study spectral forecasting or propose a new graph-wavelet backbone. Instead, it studies **post-hoc conformal prediction** for uncertainty quantification, where spectral decomposition is used only in the **second-stage calibration step** to handle non-exchangeability of graph-structured residuals. Therefore, direct comparison to prior spectral forecasting methods is not appropriate as they address a different problem.
>
> **Re: “The conformal / calibration procedure is not sufficiently explicit.”**
>
> We clarify that this procedure is described in **Sec. 3.2** and **Sec. 6.1**, following the standard **post-hoc conformal prediction** pipeline [1], [2]. Concretely, we first train a point forecaster, then calibrate lower and upper **residual quantiles** on a held-out calibration split from the decomposed residual representation, and finally add them back to the point forecast to form the prediction interval. The calibrated object is the **residual quantile** and the conformal correction is applied **after** forecasting.
>
> **Re: “The key condition behind the guarantee is not directly verified.”**
>
> We clarify that the SGCE condition constitutes a fundamental prerequisite for the coverage guarantee in Theorem 5.1, and the practical feasibility of this condition is supported across 11 real-world datasets. In detail, stronger cross-node coupling is consistently associated with worse exchangeability (Pearson **0.885**; Spearman **0.927**). Moreover, across nearly all datasets, the degree of exchangeability follows the expected ordering: it is **lowest** for the low-frequency components, **higher** for the original signal, **higher still** for the high-frequency components, and **highest** for the high-frequency components conditioned on the low-frequency structure. These results demonstrate that our design attains a near-optimal level of exchangeability required for conformal calibration.
>
> **Re: “The evaluation scope is too narrow to support the claims.”**
>
> We extend the evaluation to three non-traffic datasets, namely **AirQuality**, **Electricity**, and **Ushcn_West**, where SCALE remains competitive. More importantly, the advantage of SCALE weakens as the difference in exchangeability between the original signal and the conditioned high-frequency components falls below 0.007, and begins to disappear as this difference approaches zero. This clarifies the practical regime in which the SGCE condition empirically holds. Due to rebuttal space, we refer the reviewer to our response to Reviewer **yRod** for the full-result tables.
>
> **Re: “The approximate guarantee is not operational enough.”**
>
> Theorem 5.2 is meaningful only if its error terms admit a practical interpretation. In our view, it has clear operational significance through its two terms: $\delta_{\mathrm{leak}}$ reflects the spectral approximation error caused by imperfect decomposition, while $\delta_{\mathrm{dep}}$ reflects the effective departure from exchangeability that determines when the approximate coverage guarantee is reliable.
>
> In practice, $\delta_{\mathrm{dep}}$ is tied to cross-node coupling. Sec. 4.1 already shows that high-frequency components have much weaker correlation intensity and sparser spatial coupling than low-frequency ones. We further examined the relationship between coupling strength, measured by the correlation intensity $c_i$ in Sec. 4.1, and the exchangeability gap across 11 datasets. The correlation is strongly positive (Pearson **0.885**; Spearman **0.927**), showing that weaker coupling is consistently associated with better exchangeability. Therefore, dataset-level coupling strength provides a practical surrogate for $\delta_{\mathrm{dep}}$, helping practitioners assess in advance whether SGCE is likely to hold approximately.
>
> **Re: “The computational overhead is unclear.”**
>
> SCALE adds only a **one-time preprocessing cost** since the spectral cutoff is selected once per dataset and then reused. Runtime analysis in the setting of Table 1 shows that SCALE is **efficient**.
>
> | Method    | METR-LA (s) | PEMS 07 (s) | PEMS 08 (s) | PEMS 04 (s) |
> | --------- | ----------: | ----------: | ----------: | ----------: |
> | SCP       |        10.0 |         2.4 |         1.2 |         1.2 |
> | SeqCP     |         9.6 |        25.2 |         4.4 |         6.6 |
> | CoReL     |       843.9 |       183.5 |        46.4 |        48.2 |
> | NexCP     |      2217.2 |     10048.5 |       454.7 |       775.0 |
> | EnbPI     |     17046.0 |      6816.2 |       683.0 |      1009.3 |
> | HopCPT    |    186996.3 |    417650.7 |    272850.6 |    254422.4 |
> | ConForME  |       632.0 |      2333.0 |       338.0 |       120.8 |
> | **SCALE** |   **238.4** |   **292.0** |    **40.2** |    **99.3** |
>
> [1] Vovk et al. Algorithmic Learning in a Random World. Springer, 2005.
>
> [2] Romano et al. “Conformalized Quantile Regression.” NeurIPS, 2019.

---

> > ### Author Rebuttal · Reviewer_DHCu · 2026-04-02
> >
> > Thank you for the rebuttal. The authors clarified some of my concerns, so I am raising my score from 2 to 3.
> >
> > However, after reading the rebuttal together with the other reviews, I believe the paper would require substantial revision. The issues in writing and organization make it difficult to fully understand the core contribution and to assess whether the empirical results adequately support the claims. These issues are not likely to be resolved by small changes or additional explanation in the current version.

---

> > > ### Author Response · Authors · 2026-04-05
> > >
> > > Thank you very much for continuing to evaluate our work so carefully after reading both our rebuttal and the other reviewers’ comments. We also sincerely appreciate your recognition that the rebuttal has clarified several of your concerns, and we are grateful for your decision to raise the score.
> > >
> > > We understand your central concern at this stage: the current manuscript still has room for improvement in writing and organization, so that readers can more readily grasp the core contribution of the paper and more smoothly assess whether the empirical evidence sufficiently supports the claims. We agree that the key issue here is to present more clearly the connection among the methodological pipeline, the theoretical conditions, and the empirical conclusions.
> > >
> > > We would like to restate the contributions of our work. This paper focuses on **post-hoc conformal prediction for graph-structured multivariate time series**. More specifically, the spectral decomposition is introduced only in the calibration stage, where it is used to construct residual representations that are closer to exchangeable, thereby improving the effectiveness of conformal prediction under graph-induced non-exchangeability. This is precisely where the methodological novelty of our work lies. We also want to clarify that this paper does not propose a new spectral forecasting backbone, nor is it primarily about spectral-domain forecasting itself.
> > >
> > > At the same time, we acknowledge that, due to space constraints and the current organization of the manuscript, this main line of reasoning has not yet been presented in the most direct and accessible manner. In particular, the Preliminary section is relatively concise, which may make it harder for readers to fully follow the relationship among the method pipeline, the theoretical conditions, and the empirical regime in which the method is expected to work well. Regarding your concern about whether the empirical results adequately support the claims, we have already added cross-dataset results and analyses of the method’s boundary of validity in the rebuttal. In the final version, we will incorporate these additional results more **completely and systematically** into the main paper or appendix, so as to make clearer in which regimes the method yields consistent gains, when its advantages diminish, and how these phenomena correspond to improvements in exchangeability. We believe that these revisions will enable readers to more accurately understand the contribution of the paper and to more fully assess the extent to which the empirical findings support our claims.
> > >
> > > In the revised version, we will improve the presentation along three main directions:
> > >
> > > 1. We will make the conformal pipeline more explicit and algorithmic, clearly specifying the calibration object, the data split, the quantile computation, and the final interval construction.
> > >
> > > 2. We will distinguish more clearly among the methodological motivation, the theoretical conditions, and the empirical regime of applicability.
> > >
> > > 3. We will strengthen the theory-to-practice bridge, making it easier for readers to understand when the method is expected to be most beneficial and when its gains are likely to weaken.
> > >
> > >
> > > Overall, we sincerely appreciate your important suggestions regarding readability, organization, and the relationship between the claims and the supporting empirical evidence. We believe that the current results already demonstrate the value of the core technical idea, and that a clearer structure, more explicit presentation, and a more complete integration of the additional experiments will help readers evaluate this contribution more accurately and more thoroughly.

---

### Official Review · Reviewer_yRod · 2026-03-10

**Soundness:** 2
**Presentation:** 2
**Significance:** 2
**Originality:** 2
**Overall Recommendation:** 4
**Confidence:** 4

**Summary:**

This paper studies conformal prediction for graph-structured multivariate time series. The authors argue that cross-node coupling violates the exchangeability assumption underlying traditional conformal prediction, which can lead to unreliable uncertainty estimation. To address this issue, they propose Spectral Graph Conditional Exchangeability (SGCE) and a graph-wavelet-based framework, SCALE, which decomposes residuals into low- and high-frequency components and calibrates the high-frequency part while retaining the global trend information in the low-frequency component. Experiments on multiple real-world traffic datasets demonstrate strong empirical performance of the proposed method.

**Compliance With Llm Reviewing Policy:**

Affirmed.

**Final Justification:**

The authors’ rebuttal has addressed my previous concerns, so I have decided to raise my score from 3 to 4.

**Key Questions For Authors:**

1.Can the authors clarify whether the gains mainly come from the spectral decomposition itself or from the low-frequency conditioned gated fusion design?

2.The high-frequency branch uses only two simple statistics, STD and RMS. Have the authors compared this design with richer high-frequency representations? If so, what is the trade-off between calibration performance and model complexity?

**Limitations:**

The author did not adequately address the limitations of his research and the potential negative social impacts it might have. The authors could improve this by discussing the strong SGCE assumption, the restricted evaluation on traffic datasets, and the risks of deploying potentially miscalibrated uncertainty estimates in real-world decision-making settings.

**Strengths And Weaknesses:**

Strengths:

1.The paper is well motivated, with clear theoretical analysis and method design.

2.The experiments are comprehensive, covering different settings and necessary validations.

Weaknesses:

1.The main theoretical guarantee is built on the SGCE assumption, which appears strong. The current empirical evidence mainly shows that high-frequency components are more localized and weakly coupled; this does not directly establish exchangeability.

2.The high-frequency branch is extremely lightweight, relying only on STD and RMS statistics. While the design is motivated, the paper does not sufficiently justify whether such a minimal representation is expressive enough for calibration.

3.The experiments are limited to four traffic datasets, and additional cross-domain evaluations would be needed to better support the claimed generality.

4.The method remains a two-stage, post-hoc residual calibration framework, so its performance is still tied to the quality and distribution of residuals produced by the first-stage point forecaster.

---

> ### Author Rebuttal · Authors · 2026-03-31
>
> **Re: “The SGCE assumption seems strong, and weaker coupling does not directly prove exchangeability.”**
>
> We would like to emphasize that the high-frequency exchangeability assumption in SGCE generally holds across many real-world datasets. We verify this on **11 datasets** from various domains, where the high-frequency components are consistently much closer to the exchangeability structure required for conformal calibration. Furthermore, we quantify an empirical **exchangeability gap**, which measures the distributional change after random reordering. Across the 11 datasets, a strong positive correlation is demonstrated between **cross-node coupling strength** (measured by the correlation intensity $c_i$ defined in Sec. 4.1) and the **exchangeability gap** (Pearson **0.885**; Spearman **0.927**). This indicates that the weaker coupling is consistently associated with a higher degree of exchangeability.
>
> **Re: “Why is the high-frequency branch so lightweight? Are STD and RMS sufficient?”**
>
> The lightweight high-frequency branch is an intentional design for **robust calibration**, rather than high-capacity representation learning. Prior conformal prediction literature already shows that reliable interval calibration can be achieved by relatively simple score- or quantile-based procedures, without requiring a highly expressive second-stage model [1], [2].
>
> Under this design, STD captures fluctuation scale and RMS captures magnitude / energy, which retains the high-frequency information most relevant to interval calibration.
>
>
> **Re: “Are the gains mainly from spectral decomposition or from low-frequency conditioning / gated fusion?”**
>
> The gains come from **both spectral decomposition and low-frequency conditioning**. As already shown in the **ablation results in Table 1** and discussed in **Section 6.2 (Ablation studies)**, removing low-frequency conditioning hurts coverage, while removing spectral decomposition hurts efficiency and stability. In short, spectral decomposition provides a better calibration space, and low-frequency conditioning preserves global trend information.
>
> **Re: “The experiments are limited to traffic datasets.”**
>
> We extend our experiments to additional real-world datasets from other domains, namely **AirQuality**, **Electricity**, and **Ushcn_West**. The corresponding results, reported below, show that SCALE remains competitive beyond traffic.
>
> **AirQuality**
>
> | METHOD    | COVERAGE |    ERR |   WIDTH |     WINKLER |
> | --------- | -------: | -----: | ------: | ----------: |
> | ConForME  |   0.9372 | 0.0372 | 58.3161 |     84.4486 |
> | CoReL     |   0.9066 | 0.0066 | 40.4405 |     66.1561 |
> | NexCP     |   0.9020 | 0.0020 | 40.3555 |     71.4140 |
> | SCP       |   0.9258 | 0.0258 | 56.0753 |     83.3871 |
> | SeqCP     |   0.8791 | 0.0209 | 38.3840 |     73.0963 |
> | EnbPI     |   0.9288 | 0.0288 | 51.6976 |     78.7450 |
> | **SCALE** |   0.9124 | 0.0124 | 41.1355 | **64.3043** |
>
> **Electricity**
>
> | METHOD    | COVERAGE |    ERR |        WIDTH |       WINKLER |
> | --------- | -------: | -----: | -----------: | ------------: |
> | ConForME  |   0.9049 | 0.0049 |     770.7184 |     1238.2979 |
> | CoReL     |   0.8849 | 0.0151 |     680.6375 |     1286.3434 |
> | NexCP     |   0.8939 | 0.0061 |     777.8457 |     1160.1943 |
> | SCP       |   0.8998 | 0.0002 |     747.0611 |     1207.3663 |
> | SeqCP     |   0.8807 | 0.0193 |     746.4865 |     1160.6238 |
> | EnbPI     |   0.8901 | 0.0099 |     701.3466 |     1157.1560 |
> | **SCALE** |   0.8824 | 0.0176 | **544.8080** | **1030.5387** |
>
> **Ushcn_West**
>
> | METHOD    | COVERAGE |    ERR |       WIDTH |     WINKLER |
> | --------- | -------: | -----: | ----------: | ----------: |
> | ConForME  |   0.8976 | 0.0024 |     12.3244 |     17.1449 |
> | CoReL     |   0.8997 | 0.0003 |     11.9358 |     16.0332 |
> | NexCP     |   0.8952 | 0.0048 |     12.2161 |     16.6916 |
> | SCP       |   0.8965 | 0.0035 |     12.2692 |     17.0669 |
> | SeqCP     |   0.8774 | 0.0226 |     11.6984 |     16.9148 |
> | EnbPI     |   0.8966 | 0.0034 |     12.1393 |     16.8827 |
> | **SCALE** |   0.8964 | 0.0036 | **11.7209** | **15.8606** |
>
>
> **Re: “The method depends on the first-stage predictor.”**
>
> We agree with the reviewer, and we want to clarify that the objective of our work is to perform **conformal uncertainty quantification on top of a given forecaster**, as stated in the problem formulation (Sec. 3.1). This is also the standard setting in post-hoc conformal prediction [2], [3]. To ensure fair comparison, all methods use the same first-stage predictions and the same data split, and we also include additional first-stage backbones in **Appendix E.1** to show that the conclusion is not tied to a specific forecaster.
>
> [1] Vovk et al. Algorithmic Learning in a Random World. Springer, 2005.
>
> [2] Romano et al. “Conformalized Quantile Regression.” NeurIPS, 2019.
>
> [3] Lei et al. “Distribution-Free Predictive Inference for Regression.” JASA, 2018.

---

> > ### Author Rebuttal · Reviewer_yRod · 2026-04-03
> >
> > Thank you for the detailed rebuttal.My concerns have been addressed, and I will raise my score from 3 to 4.

---

> > > ### Author Response · Authors · 2026-04-05
> > >
> > > Thank you very much for your thoughtful feedback and for carefully considering our rebuttal. We sincerely appreciate your recognition that our response has addressed your concerns. We are also very grateful for your willingness to increase the score. Thank you again for your time, consideration, and encouraging evaluation.

---

### Official Review · Reviewer_8bHE · 2026-03-10

**Soundness:** 3
**Presentation:** 3
**Significance:** 3
**Originality:** 3
**Overall Recommendation:** 5
**Confidence:** 4

**Summary:**

This paper addresses the challenges associated with non-exchangeability in graph-structured multivariate time series (MTS) and introduces a novel Spectral Conformal Prediction method using wavelet transforms to adapt MTS for conformal prediction. By employing a decomposition strategy, the paper utilizes graph wavelets to decompose the data into low- and high-frequency components, then applies adaptive gating to conformalize the high-frequency residuals over a low-frequency embedding. Experimental results demonstrate that the proposed method not only ensures valid coverage but also consistently outperforms state-of-the-art conformal prediction methods, enhancing the coverage-efficiency trade-off.

**Compliance With Llm Reviewing Policy:**

Affirmed.

**Final Justification:**

This paper proposes an effective solution for conformal prediction in graph-structured multivariate time series. It demonstrates clear innovation, solid theoretical grounding, and follows proper writing conventions. The rebuttal process effectively addressed my concerns, which led me to revise my score. I recommend acceptance.

**Key Questions For Authors:**

Refer to Weaknesses.

**Limitations:**

The authors have not discussed the limitations and potential negative societal impacts of their work.

**Strengths And Weaknesses:**

**Strengths**

- This paper proposes an effective solution for conformal prediction in graph-structured multivariate time series.

- Theoretical analysis demonstrates that the proposed method provides a coverage guarantee.

- Experimental results show that the proposed method reduces prediction interval (PI) width by approximately 14.4%.


**Weaknesses**

- The theoretical and experimental analyses demonstrate that the high-frequency component retains sparse couplings. However, it is unclear whether the exchangeability assumption on the high-frequency component may introduce noise.

- The rationale for capturing standard deviation (STD) and root mean square (RMS) of the high-frequency component is unclear. The authors should provide a detailed explanation for this choice.

- Experiments were conducted on multiple traffic datasets, but it remains unclear whether the proposed model is applicable to data from other domains.

---

> ### Author Rebuttal · Authors · 2026-03-31
>
> **Re: “Using the high-frequency part to recover exchangeability may introduce noise.”**
>
> We agree that high-frequency residuals in real-world data may contain noise, but such noise has negligible effect on the final performance of our conformal prediction method.
>
> In this regard, prior work shows that conformal prediction is often robust to non-adversarial noise, and substantial failure mainly arises when the noise directly distorts the score distribution used for calibration [1], [2]. In our method, the high-frequency branch is not used to learn a complex predictor, but to extract lightweight window-level statistics. This makes our SCALE less sensitive to isolated spikes and avoids reintroducing strong cross-node dependence.
>
> **Re: “Why use STD and RMS?”**
>
> We use STD and RMS because they summarize the two aspects of the high-frequency components most relevant to interval adaptation, namely **fluctuation scale** and **overall magnitude**. Specifically, STD measures how strongly the high-frequency component varies within the look-back window, while RMS measures its energy level. This is also consistent with the general conformal prediction principle that interval adaptation can benefit from simple, non-learned summaries of local uncertainty [3], [4]. Therefore, we believe our compact design can already characterize the information directly related to interval estimation. We will provide a more detailed rationale in the revision.
>
> [1] Feldman et al. “Conformal Prediction is Robust to Label Noise.” ICLR, 2023.
>
> [2] Penso et al. “Estimating the conformal prediction threshold from noisy labels.” ICLR, 2025.
>
> [3] Vovk et al. Algorithmic Learning in a Random World. Springer, 2005.
>
> [4] Romano et al. “Conformalized Quantile Regression.” NeurIPS, 2019.
>
>
> **Re: “It is unclear whether the method generalizes beyond traffic datasets.”**
>
> We understand the reviewer’s concern about limited evaluation in the traffic domain. To address this concern, we conducted experiments on additional real-world datasets from non-traffic domains, including **AirQuality**, **Electricity**, and **Ushcn_West**. The new results show that SCALE remains competitive beyond traffic and continues to achieve a strong coverage-efficiency trade-off. We will incorporate these results into the revision.
>
> **AirQuality**
>
> |METHOD|COVERAGE|ERR|WIDTH|WINKLER|
> |---|---|---|---|---|
> |ConForME|0.9372|0.0372|58.3161|84.4486|
> |CoReL|0.9066|0.0066|40.4405|66.1561|
> |NexCP|0.9020|0.0020|40.3555|71.4140|
> |SCP|0.9258|0.0258|56.0753|83.3871|
> |SeqCP|0.8791|0.0209|38.3840|73.0963|
> |EnbPI|0.9288|0.0288|51.6976|78.7450|
> |**SCALE**|0.9124|0.0124|41.1355|**64.3043**|
>
> **Electricity**
>
> |METHOD|COVERAGE|ERR|WIDTH|WINKLER|
> |---|---|---|---|---|
> |ConForME|0.9049|0.0049|770.7184|1238.2979|
> |CoReL|0.8849|0.0151|680.6375|1286.3434|
> |NexCP|0.8939|0.0061|777.8457|1160.1943|
> |SCP|0.8998|0.0002|747.0611|1207.3663|
> |SeqCP|0.8807|0.0193|746.4865|1160.6238|
> |EnbPI|0.8901|0.0099|701.3466|1157.1560|
> |**SCALE**|0.8824|0.0176|**544.8080**|**1030.5387**|
>
> **Ushcn_West**
>
> | METHOD    | COVERAGE |    ERR |       WIDTH |     WINKLER |
> | --------- | -------: | -----: | ----------: | ----------: |
> | ConForME  |   0.8976 | 0.0024 |     12.3244 |     17.1449 |
> | CoReL     |   0.8997 | 0.0003 |     11.9358 |     16.0332 |
> | NexCP     |   0.8952 | 0.0048 |     12.2161 |     16.6916 |
> | SCP       |   0.8965 | 0.0035 |     12.2692 |     17.0669 |
> | SeqCP     |   0.8774 | 0.0226 |     11.6984 |     16.9148 |
> | EnbPI     |   0.8966 | 0.0034 |     12.1393 |     16.8827 |
> | **SCALE** |   0.8964 | 0.0036 | **11.7209** | **15.8606** |
>
>
> **Re: “The paper does not discuss limitations and potential negative societal impact.”**
>
>
> We thank the reviewer for pointing this out.
>
> Regarding limitations, our method relies on spectral decomposition to separate strongly coupled low-frequency components from weakly coupled high-frequency components. Its reliability may therefore be reduced on datasets where this separation is weak, residual high-frequency coupling remains strong, or the graph structure is inaccurate.
>
> Regarding societal impact, our work is methodological and does not target a specific high-risk application. However, if it is deployed in downstream decision-making systems without checking whether these structural conditions hold, the resulting prediction intervals may be miscalibrated and could lead to overconfident or misleading decisions. We therefore believe SCALE should be used together with dataset-level diagnostics and careful domain-specific validation before deployment.

---

> > ### Author Rebuttal · Reviewer_8bHE · 2026-04-03
> >
> > Thank you for the detailed response. I will raise my score from 4 to 5.

---

> > > ### Author Response · Authors · 2026-04-05
> > >
> > > Thank you very much for your kind and encouraging feedback. We sincerely appreciate your time and effort in reviewing our rebuttal, and we are glad that our responses have adequately addressed your concerns. We are also truly grateful for your willingness to increase the score. Thank you again for your thoughtful consideration and support.

---

### Decision · Program_Chairs · 2026-04-30

**Decision:**

Accept (regular)

**Comment:**

The reviewers are overall positive about this paper. The authors provide a detailed rebuttal to address reviewers’ concerns. Authors are encouraged to incorporate their response in the camera-ready.